# OViP: Online Vision-Language Preference Learning For VLM Hallucination

## Abstract

Large vision-language models (LVLMs) remain vulnerable to hallucination, often generating content misaligned with visual inputs. Although recent training-based approaches aim to mitigate hallucination, they typically rely on predefined or randomly edited negative samples that do not reflect actual model errors, thus limiting training efficacy. In this work, we propose an Online Vision-language Preference Learning (OViP) framework that dynamically constructs contrastive training data based on the model's own hallucinated outputs. By identifying semantic differences between sampled response pairs and synthesizing negative images using a diffusion model, OViP generates more relevant supervision signals in real time. This failure-driven training enables adaptive alignment of both textual and visual preferences. Moreover, we refine existing evaluation protocols to better capture the trade-off between hallucination suppression and expressiveness. Experiments on hallucination and general benchmarks demonstrate that OViP not only reduces hallucinations while preserving core multi-modal capabilities, but also substantially improves training efficiency.

## 1 Introduction

Large vision-language models (LVLMs) (Alayrac et al., 2022; Chen et al., 2023; 2024a; Liu et al., 2023; 2024b) have demonstrated remarkable performance across a wide range of multi-modal tasks (Dai et al., 2023; Li et al., 2023a; Bai et al., 2023; Wang et al., 2024b) by integrating pre-trained visual encoders with large language models (LLMs) to process and generate language grounded in visual inputs. However, LVLMs continue to struggle with persistent hallucination issues (Li et al., 2023b; Bai et al., 2024), often exhibiting incorrect references to visual content (Liu et al., 2024a; Zhou et al., 2023; Bai et al., 2024). These errors manifest as misattributing object properties, describing nonexistent entities, or fabricating spatial relationships that do not align with the image. Such inconsistencies undermine the model's faithfulness to the input and hinder further reasoning capabilities, significantly limiting the reliability of LVLMs in real-world applications.

Recent success of Direct Preference Optimization (DPO) (Rafailov et al., 2023) in LLMs alignment has inspired the exploration of multi-modal DPO to mitigate hallucination in LVLMs (Yu et al., 2024a;b; Xie et al., 2024; Sarkar et al., 2024). However, early efforts directly extend the original DPO designs from LLMs to LVLMs by constructing preference pairs solely on textual responses given the same image input, primarily focusing on response-side preference optimization and showing limited effectiveness. Recent advancements incorporate additional preference pairs conditioned on varying image inputs while keeping the same response, optimizing both visual and textual preference optimization (Wang et al., 2024a; Wu et al., 2025; Fu et al., 2025). This paradigm provides a complementary training signal that encourages the model to attend more closely to visual content.

However, prior work mainly relies on existing paired datasets (Wu et al., 2025) or expert-defined patterns to construct negative image inputs, using techniques such as random cropping (Wang et al., 2024a), noise disruption (Zhou et al., 2024a), object removal (Lu et al., 2025), or human/LLMs generated element-replaced response for image editing (Xie et al., 2024). These strategies are typically not explicitly tied to model failures, resulting in distribution misalignment between the generated negatives and the model's actual hallucination behavior, thereby offering limited improvement and failing to support adaptive and continual online[1] learning (Guo et al., 2024). Another line of re-

---

[1] We adopt the LLM community's convention of using "online" to denote "on-policy" in RL.

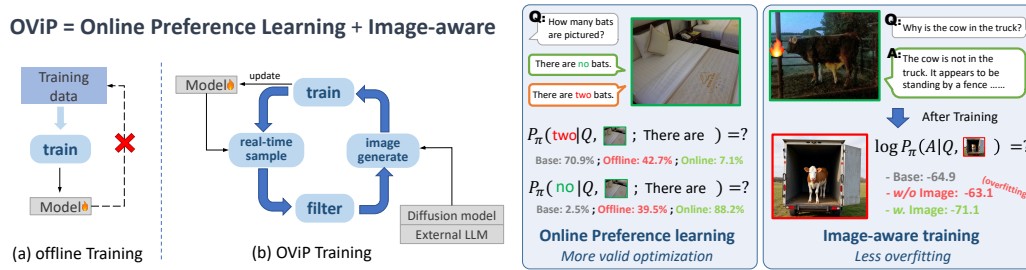

Figure 1: Offline training (a) relies on static, predefined datasets and fails to adapt to the model's evolving failure patterns, limiting its ability to address hallucinations effectively. Moreover, neglecting the role of visual input will result in overfitting to language priors. In contrast, OViP (b) combines online preference learning with image-aware training in a unified framework, enabling real-time data construction grounded in model behavior.

search explores online preference learning for hallucination mitigation, such as SIMA (Wang et al., 2025), Clip-DPO (Ouali et al., 2024b) and OPA-DPO (Yang et al., 2025). While these approaches demonstrate the importance of on-policy preference signals, their design remains limited to the textual modality and does not construct visual counterexamples on the fly. So they do not address the vision-specific failure modes of LVLMs, nor do they benefit from the proven advantages of visual preference optimization in recent multi-modal DPO studies.

Building upon these two research directions, we develop the Online Vision-Language Preference Learning Framework (OViP), a unified approach that directly targets multi-modal hallucinations through continuous, failure-aware preference learning. OViP maintains an online pipeline that first samples response pairs from the model's own outputs and then identifies their semantic differences using an external LLM. These differences guide the construction of response-conditioned negative images via a diffusion model, allowing the framework to generate both textual and visual counterexamples on the fly. By repeatedly sampling emerging failure patterns and converting them into real-time training signals, OViP adapts to the model's shifting output distribution and preserves alignment throughout training. This adaptive process expands the limited coverage of static datasets and substantially reduces the need for manual curation.

We evaluate our framework on diverse multi-modal benchmarks, covering both hallucination-focused and general tasks. Based on our experiments, we find a notable trade-off between hallucination suppression and general capability or informativeness (what we refer to as "implicit hallucination"). To address this, we refine existing evaluation protocols and reveal that many prior methods tend to overestimate their improvements. Experimental results show that OViP delivers significant advantages in both performance and efficiency. Furthermore, we investigate the role of online training and visual signals, as well as their interactions, in shaping training effectiveness.

## 2 METHODOLOGY

In this section, we first provide an overview of the Online Vision-Language Preference Learning (OViP) framework (Section 2.1). We then elaborate the process of constructing the online preference pairs during training (Section 2.2) followed by how to learn from these preference data (Section 2.3).

### 2.1 OVERVIEW

As illustrated in Figure 2, our OViP framework is designed to dynamically construct real-time preference pairs during training, by collecting in-distribution success and failure responses along with their corresponding original and synthesized images. These preference pairs are then integrated into the next training iteration for direct preference optimization on both image and response sides, providing a continuous feedback loop that refines the model's visual grounding and improves its ability to distinguish high-quality outputs from suboptimal ones.

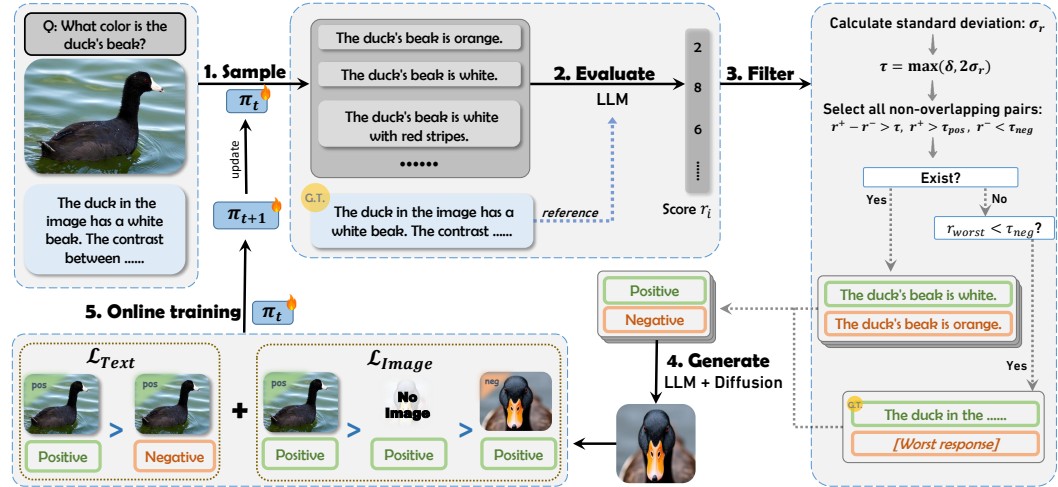

Figure 2: Overview of OViP. Given an image and a query, we employ the current model $\pi_t$ to generate multiple responses, which are then evaluated by an external LLM with reference to the ground truth. We filter and select response pairs and then generate corresponding negative images. The collected data are used to update $\pi_t$. The filtering strategy is detailed in Section 2.2.

Specifically, given an input image $\mathcal{I}^+$, an instruction $\mathcal{Q}$, and a reference response $\mathcal{A}^*$, OViP first samples multiple candidate responses using the target model $\pi$. These responses are then filtered and selected to form positive and negative pairs $(\mathcal{A}^+, \mathcal{A}^-)$. Based on the semantic discrepancies between the response pairs, contrastive images $\mathcal{I}^-$ are further synthesized to describing the negative responses. Finally, both image-level and response-level contrastive losses are applied to update the target model $\pi$. A detailed workflow of the OViP algorithm is provided in Table 6.

## 2.2 IN-DISTRIBUTION PREFERENCE DATA CONSTRUCTION

We adopt training-time inference to dynamically construct richer preference signals that continuously reflect the model's in-distribution failure modes, thereby compensating for the limited coverage of static offline datasets. At a conceptual level, the OViP framework consists of three stages: (1) real-time generation of diverse candidate outputs conditioned on visual inputs and instructions, (2) quality-aware identification of informative preference pairs that highlight the contrast between grounded and hallucinated behaviors, and (3) inverse construction of response-conditioned negative inputs that expose the model to visual evidence contradicting hallucinated outputs. We next describe the functional roles of these stages and present our practical instantiation of each component within our training pipeline.

**Real-time Generation of Output Data** At each training step $s$, given a visual input $\mathcal{I}^+$ and its corresponding textual instruction $\mathcal{Q}$, our model $\pi_s$ generates $k = 16$ candidate responses $\mathcal{A}^i$ $(i = 1, 2, \ldots, k)$ through stochastic sampling. Each generated response is then individually evaluated by an LLM-based reward function (denoted as $G_r$), which assigns a numerical reward score to each response, reflecting its alignment with the ground-truth answer $\mathcal{A}^*$.

$$\mathcal{A}^i \sim \pi_s \left( \cdot | \mathcal{I}^+, \mathcal{Q} \right); \quad r^i = G_r \left( \mathcal{A}^i, \mathcal{A}^* \right) \tag{1}$$

**Contrasting Response Pair Sampling** At this stage, OViP maintains an online pool of candidate responses for queries and identifies response pairs that exhibit meaningful quality contrast, ensuring that learning signals emphasize the distinction between successful and hallucinated behaviors, which is crucial for preference learning (Yu et al., 2025).

*In our implementation,* we dynamically construct preference pairs by selecting response pairs within each batch that display significant score disparities. Specifically, for each set of candidate responses

$\{\mathcal{A}^i\}_{i=1}^k$ with corresponding rewards $\{r^i\}_{i=1}^k$, we compute the standard deviation $\sigma_r$ of the reward scores and select pairs $(\mathcal{A}^+, \mathcal{A}^-)$ that satisfy $|r^+ - r^-| > \max(\delta, 2\sigma_r)$ where $\delta$ is a fixed lower-bound margin. This criterion ensures that only response pairs exhibiting substantial contrast in reward scores are selected, effectively emphasizing informative differences between success and failure responses. Additionally, we enforce quality constraints by requiring that the accepted positive responses meet a predefined quality criterion (i.e., $r^+ > \tau_{\text{pos}}$), while rejected negative responses fall below a specified threshold (i.e., $r^- < \tau_{\text{neg}}$). In cases where all candidate responses collectively perform poorly, we leverage offline ground-truth answers $\mathcal{A}^*$ as positive responses to guide the model learning effectively, a practice reminiscent of the mixed-policy approach in Yan et al. (2025).

$$\mathcal{D}_{\text{pair}} = \big\{ (\mathcal{Q}, \mathcal{I}^+, \mathcal{A}^+, \mathcal{A}^-) \mid \mathcal{A}^+, \mathcal{A}^- \in \{\mathcal{A}^i\}_{i=1}^k, \tag{2}$$
$$|r^+ - r^-| > \max(\delta, 2\sigma_r), r^+ > \tau_{\text{pos}}, \ r^- < \tau_{\text{neg}} \big\}$$

**Inverse Negative Image Synthesis** The goal of this stage is to construct response-conditioned negative images for visual contrastive learning. Conceptually, the framework only requires a mechanism that maps the semantic discrepancy between $(\mathcal{A}^+, \mathcal{A}^-)$ into a visually interpretable negative example, and is compatible with various instantiations such as image editing, masked manipulation, or text-to-image generation.

*In our implementation*, given a training tuple $(\mathcal{Q}, \mathcal{I}^+, \mathcal{A}^+, \mathcal{A}^-) \in \mathcal{D}_{\text{pair}}$, we synthesize negative images corresponding to negative responses while taking input images as positive. Specifically, we utilize an external LLM (denoted as $G_{\text{diff}}$) to identify a set of semantic differences between the positive and negative responses, including entities, attributes, and spatial relations, and then generate a textual description $\mathcal{T}^- = G_{\text{diff}}(\mathcal{Q}, \mathcal{A}^+, \mathcal{A}^-)$ that encapsulates the semantic content of the negative response $\mathcal{A}^-$. Subsequently, a diffusion-based image generation model (denoted as Diff) synthesizes a hard negative image as follows:

$$\mathcal{I}^- = \text{Diff}(\mathcal{T}^-) \tag{3}$$

This inverse generation process, in which the image is conditioned on the textual output, ensures that the synthesized image captures hallucinated or incorrect content, providing more targeted supervision for hallucination mitigation. Moreover, as the generation is explicitly driven by response-level discrepancies, the resulting negative images exhibit higher semantic relevance and visual specificity.

Other implementation details for training stability are provided in Appendix F.

## 2.3 IMAGE- AND RESPONSE-SIDE PREFERENCE OPTIMIZATION

To effectively align both textual and visual modalities during training, we formulate a unified optimization framework that simultaneously considers response-level and image-level preference signals. The overall optimization objective consists of two complementary components. The first is the text DPO loss(Rafailov et al., 2023), which guides the model to learn response-level preferences conditioned on the input image and instruction:

$$\mathcal{L}_{\text{Text}}\left(\mathcal{A}^+, \mathcal{A}^-; \mathcal{I}^+, \mathcal{Q}\right) = -\log\sigma\left(\beta \cdot \left[\log\frac{\pi_\theta(\mathcal{A}^+|\mathcal{I}^+, \mathcal{Q})}{\pi_{\text{ref}}(\mathcal{A}^+|\mathcal{I}^+, \mathcal{Q})} - \log\frac{\pi_\theta(\mathcal{A}^-|\mathcal{I}^+, \mathcal{Q})}{\pi_{\text{ref}}(\mathcal{A}^-|\mathcal{I}^+, \mathcal{Q})}\right]\right) \tag{4}$$

In addition to response-level alignment, we incorporate a contrastive objective focused on the visual input. By keeping the query and response fixed, the model is required to learn preferences solely from differences in the visual input. On top of this, to further ensure that the model's output maintains a reasonable and smooth probability distribution, we introduce the image-free term $\pi_\theta(\mathcal{A}|\mathcal{Q})$ and implement the image-side loss as in Wu et al. (2025):

$$\mathcal{L}_{\text{Image}}(\mathcal{I}^+, \mathcal{I}^-; \mathcal{Q}, \mathcal{A}^+) = -\log\sigma\bigg(\beta_1 \cdot \left[\log\frac{\pi_\theta(\mathcal{A}^+|\mathcal{I}^+, \mathcal{Q})}{\pi_{\text{ref}}(\mathcal{A}^+|\mathcal{I}^+, \mathcal{Q})} - \log\frac{\pi_\theta(\mathcal{A}^+|\mathcal{Q})}{\pi_{\text{ref}}(\mathcal{A}^+|\mathcal{Q})}\right]$$
$$+ \beta_2 \cdot \left[\log\frac{\pi_\theta(\mathcal{A}^+|\mathcal{Q})}{\pi_{\text{ref}}(\mathcal{A}^+|\mathcal{Q})} - \log\frac{\pi_\theta(\mathcal{A}^+|\mathcal{I}^-, \mathcal{Q})}{\pi_{\text{ref}}(\mathcal{A}^+|\mathcal{I}^-, \mathcal{Q})}\right]\bigg) \tag{5}$$

The overall loss function is then defined as:

$$\mathcal{L}_{\text{OViP}}\left(\mathcal{Q}, \mathcal{I}^+, \mathcal{I}^-, \mathcal{A}^+, \mathcal{A}^-\right) = \mathcal{L}_{\text{Text}}\left(\mathcal{A}^+, \mathcal{A}^-; \mathcal{I}, \mathcal{Q}\right) + \mathcal{L}_{\text{Image}}\left(\mathcal{I}^+, \mathcal{I}^-; \mathcal{Q}, \mathcal{A}^+\right) \tag{6}$$

| Method | Error rate | Quality | Cos Sim. |
|---|---|---|---|
| Model-based Synthesis (Ours) | 9.41% | 4.20 | 0.6224 |
| Cropping | 70.86% | 3.94 | 0.9672 |
| Random Sampling | 3.39% | 2.52 | 0.4177 |

Table 1: Quality of different images. Error Rate denotes the probability that a negative image is incorrect (not contradicting the positive response). Quality reflects GPT-based assessments of image fidelity, and Cos Sim measures the similarity between the negative image and the positive image.

## 2.4 QUALITY ANALYSIS

We evaluate both the accuracy of the LLM-based annotation in the OViP framework and the quality of the synthesized negative images.

**LLM-as-the-Annotator.** We first use GPT to evaluate the response pairs from the Contrasting Response Pair Sampling stage to check whether the preference between the positive and negative responses is unclear or even inverted. We then manually check the cases that are labeled as "incorrect". About **3.3%** of the pairs contain unclear or incorrect preference assignments. Overall, the error rate is low, indicating that using an LLM as a correctness annotator is feasible in practice.

**Negative Image Synthesis.** We further compare OViP's diffusion-based negative image synthesis strategy with two model-free negative image generation baselines used in prior works. Table 1 demonstrates that OViP's synthesized negative images exhibit higher quality and higher cosine similarity to the original images, while maintaining a low error rate.

## 3 EXPERIMENT

### 3.1 EXPERIMENTAL SETUP

**Implementation Details** We conduct our experiments on LLaVA-1.5-7B-hf and LLaVA-1.5-13B-hf (Liu et al., 2024b), with CLIP ViT-L-336px as the visual encoder and Vicuna-7b/13b as the backbone respectively. The training dataset, sourced from Yang et al. (2025), consists of 8,730 samples and 4,013 distinct image–query combinations, including image description, question answering, and some yes/no questions. We use LoRA (Hu et al., 2022) with a rank of 256 and alpha of 512. Other settings are listed in Appendix B.2

**Baselines** We compare OViP with SFT, DPO (Rafailov et al., 2023), mDPO (Wang et al., 2024a) and GRPO (Shao et al., 2024). As the original versions of SFT, DPO and mDPO are offline methods, we additionally implement iterative DPO and GRPO to facilitate a more comprehensive comparison. Furthermore, we evaluate several prior works with publicly available model weights, including HA-DPO (Zhao et al., 2023), HALVA (Sarkar et al., 2024), RLAIF-V (Yu et al., 2024b) and OPA-DPO (Yang et al., 2025). Among them, our OViP and OPA-DPO use the same original training data, which is a subset of the dataset used by RLAIF-V.

### 3.2 EVALUATION METRICS

We conduct evaluations on five **hallucination-related** and four **general capability** benchmarks to assess hallucination mitigation and overall capability degradation.

**Hallucination-Related Evaluation.** We evaluate hallucination in LVLM outputs using MMHal-Bench (MMHal) (Sun et al., 2024), AMBER generative ($AMB_{gen}$) (Wang et al., 2023), Object HalBench (ObjectHal) (Rohrbach et al., 2018), Llava-Bench-in-the-Wild (LV) (Liu et al., 2023), and AMBER discriminative ($AMB_{dis}$) (Wang et al., 2023). Detailed descriptions of the datasets, evaluation procedures, and metrics are provided in Appendix A.2

Table 2: **Main Results for OViP and other methods across different benchmarks.** The five shaded metrics highlight the key metrics for each benchmark. HRI (Hallucination Reduction Index) is the average improvement across five benchmarks. $\text{Acc}_{\text{Dif}}$ is the total accuracy changes across TextVQA(Singh et al., 2019), RealworldQA(xAI, 2024), MMStar(Chen et al., 2024b) and CVBench(Tong et al., 2024). GPT4-V([†])'s results are cited from Xiao et al. (2025),Wang et al. (2023),Duan et al. (2024) for reference. ‡ indicates the use of original evaluation strategy. * denotes methods with publicly released model weights trained on their own datasets, which we direct evaluate without re-training. ♯ signifies methods trained on datasets that are the same as or larger than ours. "2-ep" means two epochs of training. We separate offline methods from non-offline methods for clearer comparison. Detailed results of general benchmarks are provided in Appendix B.6.

| | | | $\text{AMB}_{\text{gen}}$ | | MMHal | ObjectHal | | LV | $\text{AMB}_{\text{dis}}$ | HRI | General |
| | | Chair↓ | Cover↑ | F1↑ | Score↑ | $\text{Chair}_r$ ↓ | F1↑ | Score↑ | F1↑ | | $\text{Acc}_{\text{Dif}}$ |
|---|---|---|---|---|---|---|---|---|---|---|---|
| | GPT4-V[†] | 4.6 | 67.1 | 78.8 | 3.49‡ | 13.6 | - | 95.3 | 87.4 | - | - |
| LLaVA-1.5-7B | Baseline | 7.1 | 50.0 | 65.01 | 1.90 | 51.38 | 72.40 | 57.20 | 85.5 | - | - |
| | HA-DPO* | 5.6 | 49.4 | 64.86 | 1.95 | 37.15 | 73.81 | 57.30 | 85.4 | 1.52 | *-11.59* |
| | HALVA* | 5.7 | 52.9 | **67.78** | 2.12 | 43.40 | **76.01** | 58.60 | 86.5 | **9.08** | *-7.36* |
| | RLAIF-V*♯ | 3.1 | 49.8 | 65.79 | 2.54 | 9.35 | 69.78 | 58.90 | 86.4 | 1.37 | *-6.74* |
| | OPA-DPO*♯ | 2.4 | 45.2 | *61.79* | **2.78** | 6.37 | *63.26* | **64.80** | 86.7 | *-5.60* | *-11.82* |
| | SFT | 3.5 | 50.6 | 66.39 | 2.52 | 20.60 | 70.30 | *52.20* | 86.1 | -1.47 | *-8.07* |
| | DPO | 3.7 | 48.9 | 64.86 | 2.35 | 26.60 | 71.95 | 56.70 | 86.8 | 1.65 | -3.86 |
| | mDPO | 3.4 | 48.6 | 64.67 | **2.55** | 25.45 | **73.92** | 55.80 | 86.1 | 2.99 | -3.05 |
| | $\text{DPO}_{\text{iterative}}$ | 3.9 | 48.7 | 64.64 | 2.32 | 27.11 | 72.33 | 56.40 | 86.5 | 1.31 | -2.98 |
| | $\text{GRPO}_{\text{2ep}}$ | 4.8 | 51.2 | 66.59 | 2.45 | 34.98 | 73.83 | 58.70 | 86.8 | 6.75 | -3.83 |
| | **OViP** | 4.0 | 51.1 | **66.70** | 2.52 | 33.22 | 73.50 | **63.10** | 87.3 | 9.58 | **+0.88** |
| | $\text{OViP}_{\text{2ep}}$ | 4.0 | 51.6 | **67.12** | **2.65** | 29.54 | **74.18** | 60.90 | 87.4 | 10.00 | -1.01 |
| LLaVA-1.5-13B | Baseline | 6.5 | 51.0 | 65.99 | 2.24 | 46.18 | 76.73 | 62.60 | 89.1 | - | - |
| | HALVA* | 6.0 | 52.2 | 67.12 | 2.45 | 35.07 | **77.75** | 61.70 | 90.0 | 4.22 | *-5.45* |
| | OPA-DPO*♯ | 2.8 | 47.8 | 64.08 | **2.88** | 5.88 | *64.46* | 64.70 | 89.3 | *-7.05* | -9.66 |
| | SFT | 4.5 | 50.0 | 65.64 | 2.38 | 31.21 | 75.81 | 64.00 | 89.9 | 1.79 | -1.24 |
| | DPO | 3.6 | 50.6 | 66.37 | 2.53 | 25.00 | 75.00 | 65.30 | 89.6 | 2.42 | +0.12 |
| | mDPO | 3.9 | 50.1 | 65.86 | 2.51 | 21.79 | 75.35 | 64.50 | 89.5 | 1.78 | -1.12 |
| | $\text{GRPO}_{\text{2ep}}$ | 3.8 | 52.4 | 67.84 | 2.38 | 23.76 | 75.55 | **66.70** | 90.4 | 4.96 | -1.48 |
| | **OViP** | 4.4 | 53.1 | **68.28** | **2.58** | 36.30 | 76.52 | 64.60 | 89.7 | 5.25 | **+0.85** |
| | $\text{OViP}_{\text{2ep}}$ | 3.6 | 53.7 | **68.98** | 2.57 | 28.62 | **76.75** | **67.90** | 90.2 | 8.02 | **+2.02** |

Prior work has primarily focused on assessing the precision of model outputs, i.e., whether the generated content contains explicit hallucinations. However, this perspective often overlooks the *completeness* of the output: a model may omit relevant entities (especially in image description tasks), leading to what we term *implicit hallucinations*. *We argue that both explicit and implicit hallucinations are critical for a faithful evaluation of model reliability.* Building on this perspective and the observation of failure cases where existing benchmarks can be hacked, we **refine the evaluation protocols and introduce an F1 score for $\text{AMB}_{\text{gen}}$ and ObjectHal to better capture the extent of hallucination in generated responses**. Illustrative failure cases of prior evaluation strategies are presented in Appendix A.3.

To aggregate performance across five benchmarks, we introduce the **Hallucination Reduction Index** (**HRI**) as a unified measure of overall improvement. HRI is computed by summing the normalized improvements from each benchmark to obtain the overall relative gain. The detailed calculation of HRI and the discussion of its fairness are provided in Appendix A.1.

**General Capability Evaluation** To assess the trade-off between hallucination mitigation and general visual capability, we evaluate the trained models on general benchmarks, including RealworldQA (xAI, 2024), TextVQA (Singh et al., 2019), CVBench (Tong et al., 2024), MMStar (Chen et al., 2024b). We aggregate the results across these benchmarks and compute the **Accuracy Difference**, serving as a unified metric to quantify overall performance variation after training.

## 3.3 MAIN RESULTS

Table 2 presents results for OViP and other methods across multiple benchmarks on various LVLM backbones. **OViP consistently achieves significant improvements across most primary metrics while effectively preserving the model's general visual capabilities** (achieving +0.88 with one

epoch for General $\text{Acc}_{\text{Dif}}$ and a slight drop of -1.01 for 2 epochs), whereas most other methods that exhibit varying degrees of degradation in general benchmarks. Moreover, OViP further improves with an additional training epoch. Notably, even with one epoch, OViP surpasses HALVA and 2-epoch GRPO, both of which utilize twice as much training data, but still yield lower HRI and suffer from general ability degradation.

A critical phenomenon often overlooked in previous work (Xie et al., 2024; Yang et al., 2025; Fu et al., 2025; Xiao et al., 2025; Wang et al., 2024a; Yu et al., 2024a;b) is that **offline methods generally impair models' general capability while also introducing implicit hallucinations** (as discussed in subsection 3.2). This issue is particularly evident in OPADPO, where Chair score on $\text{AMB}_{\text{gen}}$ drops to 2.4, and Cover metric decreases from the initial 50.0 to 45.2, far below other methods. An illustrative example of such omission is in Figure 8 in Appendix. Moreover, excessive training further exacerbates this problem: as shown in Table 2, several DPO-like methods (HA-DPO, HALVA, RLAIF-V, OPA-DPO) trained for more than two epochs suffer from much larger declines in general capability compared to DPO and mDPO trained for only one epoch. At the same time, except for HALVA, their HRI scores are also lower than those of DPO and mDPO, which mainly influenced by the low F1 scores on $\text{AMB}_{\text{gen}}$ and ObjectHal. ==With these possible signs of overfitting, we suggest that some improvements reported in prior work may be overestimated.==

## 3.4 ABLATION STUDY

**Loss functions.** We evaluated various combinations of loss functions for online preference learning in hallucination mitigation to derive the final formulation in Equation 6. Our ablation study examines the effectiveness of different training objectives, including text-side ($\mathcal{L}_{\text{Text}}$), image-side and auxiliary losses. Specifically for image-side losses, we examine our image loss $\mathcal{L}_{Image}$ alongside two variants $\mathcal{L}_{Image}^{base}$ and $\mathcal{L}_{Image-Sym}$. For auxiliary loss, we compare the anchor loss proposed by Wang et al. (2024a) and the bidirectional anchor loss, which enforce the probability of positive response to increase and the negative one to decrease. Detailed formulations are provided in Appendix B.1. ==The ablation for auxiliary loss is provided in Appendix== B.3.

Table 3: Results of different loss functions. $\mathcal{L}_{\text{OViP}} = \mathcal{L}_{\text{Text}} + \mathcal{L}_{\text{Image}}$.

| Loss Functions | HRI | |
|---|---|---|
| | *From Scratch* | *Iterative* |
| $\mathcal{L}_{\text{OViP}}$ | **4.32** | **7.94** |
| $- \mathcal{L}_{\text{Text}}$ | 4.23 | 7.71 |
| $- \mathcal{L}_{\text{Image}}$ | -2.29 | 4.56 |
| $\mathcal{L}_{\text{Text}} + \mathcal{L}_{\text{Image}}^{base}$ | 4.08 | 7.50 |
| $\mathcal{L}_{\text{Image-Sym}}$ | -0.32 | 6.57 |

Table 4: Results of offline and online training strategy with DPO and OViP. Cover measures the informativeness of the model from $\text{AMB}_{\text{gen}}$. Cover score of the original model is 50.

| Method | | Cover | HRI | General $\text{Acc}_{\text{Dif}}$ |
|---|---|---|---|---|
| OViP | online | **51.1** | **9.36** | **0.88** |
| | offline | 49.9 | 4.32 | 0.08 |
| DPO | online | **50.0** | **1.71** | -2.57 |
| | offline | 48.3 | -2.29 | **-1.38** |

We conduct experiments under two training regimes: (1) training from scratch, and (2) iterative training initialized with a DPO-pretrained model using the existing dataset, to ablate these losses on top of different initialized models with varying capabilities. We observe that models trained with different losses do not suffer from a notable drop in general ability (**General $\text{Acc}_{\text{Dif}} > -1.5$**). Therefore, in Table 3 we only report the HRI results, which show that the full OViP loss consistently outperforms all variants under both training regimes. Moreover, the form of the image loss greatly affects the results, with the loss in Equation 5 achieving the best performance.

**Online v.s. Offline.** Table 4 demonstrates that online training consistently outperforms its offline counterpart in HRI by at least 4 points within just one epoch (and continues to improve with further training, while offline training suffers from overfitting). Another notable observation is that *online training also improves the informativeness of model outputs*. Even when trained with DPO, the Cover score remains 50. In contrast, previous studies (Yu et al., 2024b; Yang et al., 2025; Fu et al., 2025) using the similar dataset typically exhibit a drop in this aspect. Additionally, the improvement for online training over offline training is almost across every individual benchmark and each corresponding metric, online training yields more stable and superior performance. Detailed results are provided in Appendix B.4.

| Method | AMB$_{gen}$ | | | MMHal | ObjectHal | | LV | AMB$_{dis}$ | General |
| | Chair | Cover | **F1** | **Score** | Chair$_r$ | **F1** | **Score** | **F1** | **Acc$_{Dif}$** |
|---|---|---|---|---|---|---|---|---|---|
| Cropping | 3.5 | 52.50 | 68.00 | 2.50 | 27.21 | 73.75 | 62.50 | 86.40 | +0.34 |
| Sampling | 3.4 | 50.50 | 66.33 | **3.37** | **24.20** | 73.11 | 61.20 | **86.60** | -0.66 |
| **Synthesis** | **3.3** | **52.80** | **68.30** | 2.70 | 28.21 | **74.14** | **63.60** | 85.70 | **+1.44** |

Table 5: Results for different negative image generation methods. **Cropping** refers to randomly removing 0–20% of the positive image. **Sampling** denotes selecting a random image from the entire dataset as the negative image. **Synthesis** corresponds to the diffusion-based negative image generation method used in OViP.

**Negative Images.** We further investigate how different strategies for generating negative images affect model performance. As shown in Table 5, we compare three approaches: cropping (adopted in mDPO)(Wang et al., 2024a), random sampling, and model-based online synthesis. The images obtained through Random Sampling have low semantic relevance to the original text and do not lie in the model's hallucination distribution. This approach suffers from the same informativeness degradation observed in offline methods in Cover of AMBgen and F1 of ObjectHal are low. In contrast, the Cropping strategy does not incur such losses in informativeness or general capability, likely because the cropped images still preserve partial semantic alignment. Overall, the diffusion-based online synthesis used in OViP delivers the most favorable performance, effectively generating high-quality hallucination-targeted negative images.

## 4 FURTHER STUDY

### 4.1 TRAINING EFFICIENCY

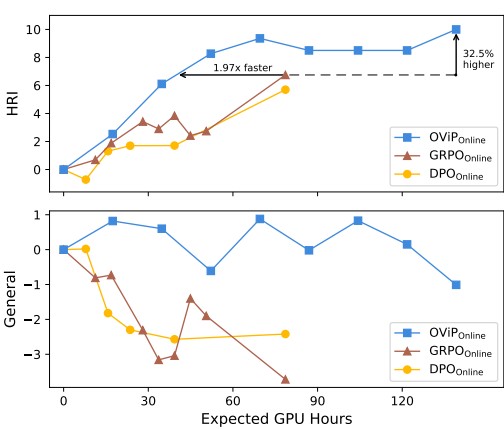
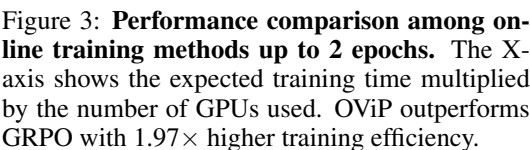

Figure 3: **Performance comparison among online training methods up to 2 epochs.** The X-axis shows the expected training time multiplied by the number of GPUs used. OViP outperforms GRPO with 1.97× higher training efficiency.

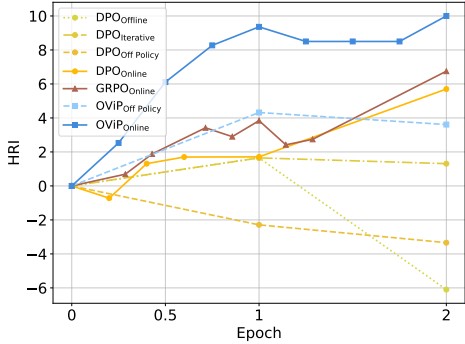

Figure 4: **Results under different training strategies**. Offline denotes training with the existing dataset; Off Policy refers to training with sampled data; and Iterative indicates that the dataset for the second epoch is generated by sampling from the offline-trained model after the first epoch.

Although OViP requires constructing negative images, which needs additional GPU resources for deploying diffusion models and incurs extra time overhead, we show that OViP still has clear advantages in overall training efficiency. In Figure 3, we compare different online methods by plotting their HRI and general capability against expected GPU hours. (A detailed analysis of training cost and efficiency is provided in Appendix D) *The results show that despite slower per-iteration speed, OViP achieves approximately 1.97× higher training efficiency than GRPO.* OViP requires only about half the computation of GRPO to achieve comparable performance, while Online DPO

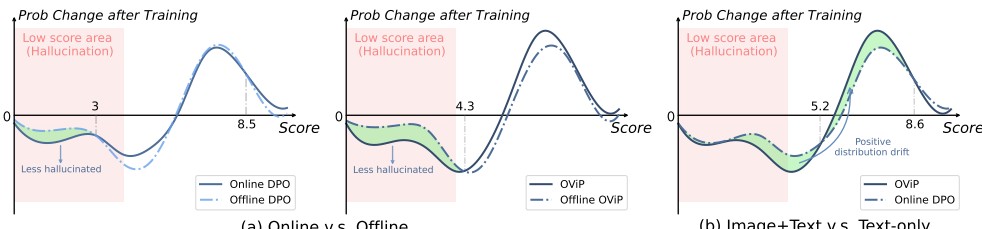

Figure 5: Change in probability mass for the responses with corresponding score after training. We smooth the discrete probability changes over 11 score bins (0-10) into a continuous curve. "Low score" refers to scores less than 4. "Change > 0" represents the probability increases after training.

performs slightly worse than GRPO. As for offline approaches, although their data construction and training require a similar amount of computation, their performance consistently falls short of their online counterparts; hence, our efficiency comparison focuses on online methods.

## 4.2 TRAINING DYNAMICS

Figure 4 illustrates how HRI evolves during training under different strategies, which allows us to investigate the dynamics of hallucination throughout training.

**Need for Visual and Online Signals**   For hallucination mitigation in LVLMs, *adding visual supervision signals proves crucial*: offline OViP surpasses GRPO and all DPO variants with one epoch. Building on this, *online methods offer further advantages, which not only make each optimization step more effective in reducing hallucinations, but also exhibit better scalability*, with overfitting arising significantly later compared to non-online approaches, whose performance starts to drop after training for one epoch. We conjecture that this superiority stems from the model-specific nature of hallucinations, which requires supervision to precisely target the current model's errors.

**Early Training Stagnation**   Both Online DPO and Off-Policy DPO exhibit an initial drop in performance, while GRPO and OViP show relatively slow improvement during the early stages of training. We attribute this phenomenon to the model's initially skewed output distribution. Early training primarily increases the diversity of model outputs, which does not immediately translate into performance gains but expands the search space for subsequent learning. A detailed discussion is provided in Appendix B.5.

## 4.3 WHAT DO WE ACTUALLY OPTIMIZE DURING TRAINING?

To understand how different training strategies reshape the model's behavior, we sample responses, score them using our evaluator, and track the change in probability mass across score levels before and after training(Figure 5). This analysis reveals how each method redistributes probability over the actual responses the model would generate.

**Online preference learning suppresses severe hallucinations more effectively than offline DPO.** Offline DPO barely shifts probability mass away from extreme failures (scores < 4). In contrast, online training continuously exposes the model to its own most confident mistakes, allowing the model to substantially decrease the probability of severely hallucinated outputs.

**Visual preference learning induces broad quality improvements but offers limited gains for extreme hallucinations.**   Adding image-based supervision consistently shifts probability mass upward across mid-quality responses, indicating better grounding and more informative answers. However, it does not significantly further suppress the lowest-scoring outputs.

**These two components influence different regions of the response-quality distribution.**   Online learning primarily corrects severe, high-confidence hallucinations, whereas visual preference learn-

==highlight== ing improves overall grounding and informativeness. Their effects are therefore complementary: one targets the left tail of the distribution, the other lifts the central and right regions. This complementary behavior explains why combining the two yields stable, additive improvements across all evaluation metrics. ==highlight==

## 5 RELATED WORK

### 5.1 LVLM HALLUCINATION

Works of synthetic data construction for mitigating hallucination in LVLMs can be broadly categorized into image-related synthesis and text-only synthesis. On the image side, several approaches leverage entity extraction and masking to perform targeted image editing, generating visually similar but semantically distinct counterfactuals (Xie et al., 2024; Lu et al., 2025). In contrast, Hallusion-Bench (Guan et al., 2024) adopts a manual approach, carefully crafting counterfactual images to probe specific failure modes. Other works take a generative perspective: SynthVLM (Liu et al., 2024c) and SynthEmbedding (Sharifzadeh et al., 2024) utilize off-the-shelf models to synthesize new images or directly generate image embeddings for hallucination-aware training. Meanwhile, text-side data augmentation can also be used in LVLM training. VoCoT (Li et al., 2024) introduces new prompting patterns and response types to generate hallucination-prone QA data at scale. Other works such as Zhou et al. (2024a), Sarkar et al. (2024), Amirloo et al. (2024) introduce noise through perturbation, masking, or controlled corruption to simulate erroneous responses. More recent approaches (Xiao et al., 2025; Yu et al., 2024a) aim to detect and correct hallucinated content at varying levels of granularity, from token-level edits to full-sequence rewrites.

These efforts significantly improve the diversity and coverage of supervision signals available for training hallucination-robust VLMs.

### 5.2 ALLOCATING MORE COMPUTATION ON TRAINING SAMPLE CONSTRUCTION

Recent research has increasingly adopted the paradigm of allocating additional computation during training to get better training samples. Several studies utilize reinforcement learning with human or AI-generated feedback to guide VLM outputs. RLHF-V (Yu et al., 2024a) leverages fine-grained human annotations to correct hallucinated content, while RLAIF-V (Yu et al., 2024b) replaces human labels with feedback from ensembles of open-source models, significantly reducing annotation overhead. Similarly, OPA-DPO (Yang et al., 2025) employs an on-policy editing step prior to DPO, aligning training samples closely with model predictions to enhance data efficiency. CLIP-based methods dynamically filter self-generated samples for high-quality training pairs (Ouali et al., 2024a; Zhou et al., 2024b). Other methods integrate auxiliary reward models or evaluators during training, providing continuous and adaptive feedback loops (Sun et al., 2024; Yan et al., 2024). Additionally, recent approaches incorporate reasoning or editing mechanisms directly into training, using iterative self-feedback or generative data augmentation techniques to dynamically refine model outputs (Zhao et al., 2023; Kim et al., 2024). These strategies improve model alignment and factuality by enriching the quality and relevance of supervision signals during training.

## 6 CONCLUSION

In this work, we propose the Online Vision-language Preference Learning (OViP) framework to efficiently address the hallucination problem in LVLMs. By integrating online preference learning with image-aware training, OViP enables real-time construction of high-quality contrastive data during training. Furthermore, to better assess the trade-offs between hallucination reduction and overall performance, we refine and extend existing evaluation protocols. Experimental results demonstrate that OViP significantly outperforms prior offline/online training approaches, achieving substantial hallucination reduction while preserving general vision-language capabilities, which many existing offline methods fail to preserve. Our investigation into training dynamics also sheds light on the underlying mechanisms behind OViP's effectiveness.

ETHICS STATEMENT

This work focuses on improving the factual reliability of vision-language models by reducing hallucination. While it does not directly engage with societal applications, it contributes to the broader goal of building more trustworthy and robust AI systems. Although the method itself does not pose obvious risks, we note that even improved generation quality does not eliminate the possibility of misuse, such as producing misleading content. Responsible deployment and proper safeguards remain necessary when integrating such models into real-world applications.

REPRODUCIBILITY STATEMENT

We provide detailed descriptions of the training and evaluation setups in Appendix A and Appendix B. In addition, we include anonymized training and evaluation code, instructions for running the experiments, and information on accessing the relevant datasets in the supplementary materials.

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

## A  EVALUATION

### A.1  HALLUCINATION REDUCTION INDEX

#### A.1.1  METRIC DESIGN

HRI represents an aggregate improvement metric across five different benchmarks. Simply summing the raw scores from each benchmark would not be a reasonable or rigorous approach, as the metrics are not directly comparable. Therefore, we calculate the improvement ratio for each benchmark based on its potential improvement range, effectively converting the raw metric gains into an additive proportion of improvement. Furthermore, we employ a conservative aggregation method to avoid overestimating the effectiveness of our approach.

Let $a_i, i \in \{1, 2, 3, 4, 5\}$ denotes $\text{F1}_{\text{AMB}-\text{gen}}, \text{Score}_{\text{MMHal}}, \text{F1}_{\text{ObjectHal}}, \text{LV}_{\text{score}}, \text{F1}_{\text{AMB}-\text{dis}}$ respectively, namely the results on each benchmark, superscript "base" represents performances of the baseline model and "ref" represents the set reference performances. Then HRI is calculated as:

$$\textbf{HRI} = 2 \times \sum_{i=1}^{5} \frac{a_i - a_i^{\text{base}}}{a_i^{\text{ref}} - a_i^{\text{base}}} \tag{7}$$

#### A.1.2  MAIN RESULTS

For 7B model, we set the reference performances as $\textbf{OViP}_{2\text{ep}}$, so it comes:

$$\textbf{HRI} = 2 \times \left( \frac{a_1 - 65.01}{67.12 - 65.01} + \frac{a_2 - 1.90}{2.65 - 1.90} + \frac{a_3 - 72.40}{74.18 - 72.40} + \frac{a_4 - 57.20}{60.90 - 57.20} + \frac{a_5 - 85.5}{87.4 - 85.5} \right)$$

For 13B model, we also use $\textbf{OViP}_{2\text{ep}}$ as the reference performances except for the ObjectHal benchmark which almost all methods fail to improve. We set the reference performance of ObjectHal to 79.0.

$$\textbf{HRI} = 2 \times \left( \frac{a_1 - 65.99}{68.98 - 65.99} + \frac{a_2 - 2.24}{2.57 - 2.24} + \frac{a_3 - 76.73}{79.00 - 76.73} + \frac{a_4 - 62.60}{67.90 - 62.60} + \frac{a_5 - 89.1}{90.2 - 89.1} \right)$$

#### A.1.3  ABLATION STUDY: LOSS FUNCTIONS

There is no method surpassing other methods significantly, so we consider the best performance on the benchmark as its reference peerformance.

$$\textbf{HRI} = 2 \times \left( \frac{a_1 - 65.01}{68.57 - 65.01} + \frac{a_2 - 1.90}{2.70 - 1.90} + \frac{a_3 - 72.40}{74.14 - 72.40} + \frac{a_4 - 57.20}{64.10 - 57.20} + \frac{a_5 - 85.5}{87.20 - 85.5} \right)$$

#### A.1.4  ABLATION STUDY: ONLINE AND OFFLINE

Same as Main Results.

$$\textbf{HRI} = 2 \times \left( \frac{a_1 - 65.01}{67.12 - 65.01} + \frac{a_2 - 1.90}{2.65 - 1.90} + \frac{a_3 - 72.40}{74.18 - 72.40} + \frac{a_4 - 57.20}{60.90 - 57.20} + \frac{a_5 - 85.5}{87.4 - 85.5} \right)$$

#### A.1.5  FURTHER STUDY

Same as Main Results.

$$\textbf{HRI} = 2 \times \left( \frac{a_1 - 65.01}{67.12 - 65.01} + \frac{a_2 - 1.90}{2.65 - 1.90} + \frac{a_3 - 72.40}{74.18 - 72.40} + \frac{a_4 - 57.20}{60.90 - 57.20} + \frac{a_5 - 85.5}{87.4 - 85.5} \right)$$

#### A.1.6  FAIRNESS

When aggregating different metrics through weighted averaging, it is necessary to account for the relative importance of each metric. Here, we define the potential improvement of a metric by considering its maximum observed gain in comparable experiments, and assign its weight as the inverse

of this gain to normalize across metrics. For example, if metric A shows a maximum improvement of 2 points while metric B improves by 4 points, we assume that an equally strong model would, on average, achieve only half as much gain on A as on B. Consequently, each point of improvement on A should be considered twice as important as a point on B. Compared with simple averaging, this weighting scheme better reflects the relative significance of different metrics and is less prone to being gamed.

## A.2 BENCHMARKS

- MMHal-Bench (MMHal) (Sun et al., 2024) is a model-evaluated question-answering benchmark covering 8 categories and 12 topics. While the original evaluation strategy uses `GPT-4` to judge model responses, a text-only model will introduce considerable judging-time hallucinations and errors, so `gpt-4o-2024-05-13` is better for evaluation. (Amirloo et al., 2024).

- AMBER generative ($AMB_{gen}$) (Wang et al., 2023) is a judging-model-free benchmark for the image description task, comprising 1,004 samples. **Chair** measures the object-level hallucination rate as the average precision of objects mentioned in the model's descriptions, while **Cover** indicates the recall of objects. *We observe a noticeable trade-off between these two metrics across various methods, where improvements in one often come at the expense of the other. To provide a more balanced and overall assessment, we introduce a new **F1** score calculated as the harmonic mean of Chair and Cover.*

- Object HalBench (ObjectHal) (Rohrbach et al., 2018) evaluates object-level completeness and hallucination rates. The generation prompts are augmented from Yu et al. (2024a). **Chair$_r$** denotes the response-level hallucination rate. We also introduce an object-level **F1** metric to comprehensively measures the balance between hallucination and object coverage. Objects extraction is performed using `gpt-4o-2024-05-13`.

- Llava-Bench-in-the-Wild (LV) (Liu et al., 2023) evaluates models' visual abilities, using 60 open-ended questions grounded in 24 diverse images from real-world and abstract scenarios. The evaluation is conducted using `gpt-4o-2024-05-13`.

- AMBER discriminative ($AMBER_{dis}$) (Wang et al., 2023) includes 14,216 'Yes/No' questions regarding objects in image. We use the **F1** score as its metrics.

## A.3 BAD CASES

### A.3.1 MMHAL

Shown in Figure 6, the original evaluation protocol utilizes the text-only `gpt-4-turbo-2024-04-09` to evaluate the model response, which has no access to the input image and can only infer from the given image contents and ground truth, so it will lead to many incorrect judgments. We replace it with `gpt-4o-2024-05-13`, which yields more accurate assessments.

### A.3.2 AMBER-GENERATIVE & OBJECTHAL

AMBER uses an automatic method for detecting the hallucination entity, which primarily relies on the pre-defined hallucination words. ObjectHal introduces LLM to extract the mentioned entities, its metrics are basically the same with AMBER. Figure 7 illustrates several cases of misjudgment in AMBER. Since the score is determined solely by the presence of specific predefined words rather than the actual semantic correctness, the hallucination rate (Chair score) is often overestimated. **Moreover, this issue becomes more pronounced as the diversity and informativeness of model responses increases.**

Many methods achieve great improvements in the Chair score (entity-wise hallucination rate), but often at the cost of a significant decrease in the cover rate (completeness and informativeness). Figure 8 provides an example of this information deficit phenomenon, which should also be considered in the evaluation of model performance.

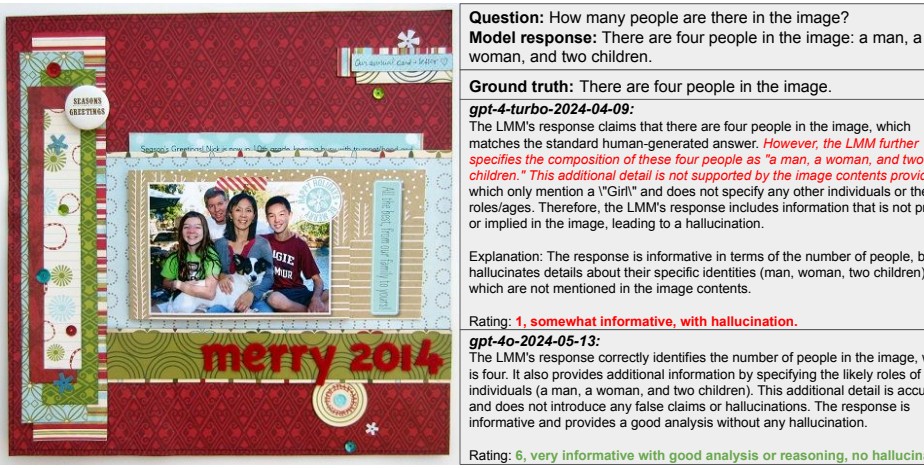

Figure 6: Text-only LLM can not correctly judge the response.

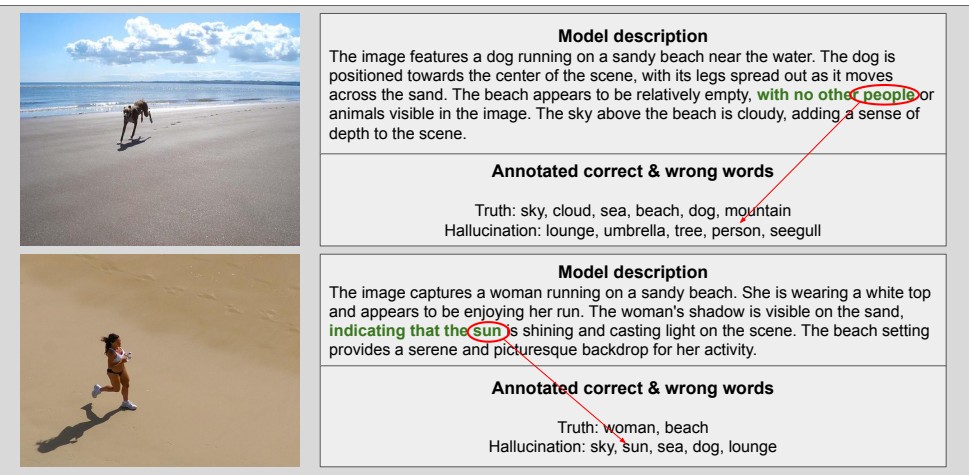

Figure 7: Rule-based extraction will lead to misjudgments to some extent.

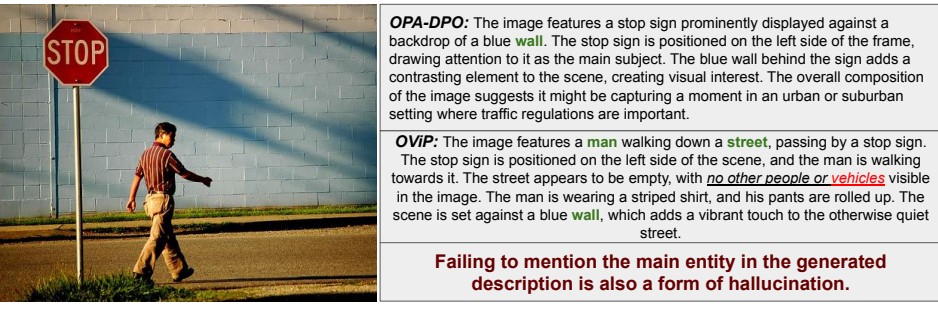

Figure 8: OPA-DPO fails to mention the man, a deficiency that is captured by Cover score but often overlooked in previous evaluations. "vehicles" is incorrectly identified as a hallucination word.

## B  EXPERIMENTS

### B.1  LOSS FUNCTIONS

Base image loss $\mathcal{L}_{\text{Image}}^{base}$ is similar to DPO loss which replace the response pair with the image pair:

$$\mathcal{L}_{\text{Image}}^{base}\left(\mathcal{I}^+, \mathcal{I}^-; \mathcal{Q}, \mathcal{A}^+\right) = \log \sigma \left( \beta \cdot \left[ \log \frac{\pi_\theta(\mathcal{A}^+|\mathcal{I}^+, \mathcal{Q})}{\pi_{\text{ref}}(\mathcal{A}^+|\mathcal{I}^+, \mathcal{Q})} - \log \frac{\pi_\theta(\mathcal{A}^+|\mathcal{I}^-, \mathcal{Q})}{\pi_{\text{ref}}(\mathcal{A}^+|\mathcal{I}^-, \mathcal{Q})} \right] \right)$$

Symmetrical image loss $\mathcal{L}_{\text{Image}-\text{Sym}}$ considers the negative image and the negative response a correct pair, then calculate Image loss using negative response and image as the positive one:

$$\begin{aligned}
\mathcal{L}_{\text{Image}-\text{Sym}}\left(\mathcal{I}^+, \mathcal{I}^-, \mathcal{A}^+, \mathcal{A}^-; \mathcal{Q}\right) = \ & \mathcal{L}_{\text{Image}}(\mathcal{I}^+, \mathcal{I}^-; \mathcal{Q}, \mathcal{A}^+) + \mathcal{L}_{\text{Image}}(\mathcal{I}^-, \mathcal{I}^+; \mathcal{Q}, \mathcal{A}^-) \\
= \ & -\log \sigma \left( \beta_1 \cdot \left[ \log \frac{\pi_\theta(\mathcal{A}^+|\mathcal{I}^+, \mathcal{Q})}{\pi_{\text{ref}}(\mathcal{A}^+|\mathcal{I}^+, \mathcal{Q})} - \log \frac{\pi_\theta(\mathcal{A}^+|\mathcal{Q})}{\pi_{\text{ref}}(\mathcal{A}^+|\mathcal{Q})} \right] \right. \\
& \left. + \beta_2 \cdot \left[ \log \frac{\pi_\theta(\mathcal{A}^+|\mathcal{Q})}{\pi_{\text{ref}}(\mathcal{A}^+|\mathcal{Q})} - \log \frac{\pi_\theta(\mathcal{A}^+|\mathcal{I}^-, \mathcal{Q})}{\pi_{\text{ref}}(\mathcal{A}^+|\mathcal{I}^-, \mathcal{Q})} \right] \right) \\
& -\log \sigma \left( \beta_1 \cdot \left[ \log \frac{\pi_\theta(\mathcal{A}^-|\mathcal{I}^-, \mathcal{Q})}{\pi_{\text{ref}}(\mathcal{A}^-|\mathcal{I}^-, \mathcal{Q})} - \log \frac{\pi_\theta(\mathcal{A}^-|\mathcal{Q})}{\pi_{\text{ref}}(\mathcal{A}^-|\mathcal{Q})} \right] \right. \\
& \left. + \beta_2 \cdot \left[ \log \frac{\pi_\theta(\mathcal{A}^-|\mathcal{Q})}{\pi_{\text{ref}}(\mathcal{A}^-|\mathcal{Q})} - \log \frac{\pi_\theta(\mathcal{A}^-|\mathcal{I}^+, \mathcal{Q})}{\pi_{\text{ref}}(\mathcal{A}^-|\mathcal{I}^+, \mathcal{Q})} \right] \right)
\end{aligned}$$

Anchor loss $\mathcal{L}_{\text{Anchor}}$ directly enforces the probability of positive response to be higher for intuitively better optimization results.

$$\mathcal{L}_{\text{Anchor}}(\mathcal{A}^+, \mathcal{A}^-; \mathcal{Q}, \mathcal{I}^+) = -\log \sigma \left( \beta \cdot \log \frac{\pi_\theta(\mathcal{A}^+|\mathcal{I}^+, \mathcal{Q})}{\pi_{\text{ref}}(\mathcal{A}^+|\mathcal{I}^+, \mathcal{Q})} \right)$$

Bi-directional anchor loss $\mathcal{L}_{\text{Bi}-\text{Anchor}}$ not only exerts supervision on the positive response, but it also makes the negative response probability to be lower.

$$\mathcal{L}_{\text{Bi}-\text{Anchor}}(\mathcal{A}^+, \mathcal{A}^-; \mathcal{Q}, \mathcal{I}^+) = -\log \sigma \left( \beta \cdot \log \frac{\pi_\theta(\mathcal{A}^+|\mathcal{I}^+, \mathcal{Q})}{\pi_{\text{ref}}(\mathcal{A}^+|\mathcal{I}^+, \mathcal{Q})} \right) + \log \sigma \left( \beta \cdot \log \frac{\pi_\theta(\mathcal{A}^-|\mathcal{Q})}{\pi_{\text{ref}}(\mathcal{A}^-|\mathcal{Q})} \right)$$

### B.2  SETTINGS

By default, we use the following settings:

**Software infrastructure.**  In our implementation, we deploy the non-training LLM and diffusion models as services using FastAPI. During training, the system interacts with these services via API calls to obtain feedback, image prompts, and the paths to generated images.

**Models.**  The LLM we use for judging response and providing image-generation prompt is `Qwen-2.5-7b-instruct` (https://huggingface.co/Qwen/Qwen2.5-7B-Instruct). The diffusion model for image generation is `FLUX.1-dev` (https://huggingface.co/black-forest-labs/FLUX.1-dev).

**Training**  Both the 7B and 13B models are trained for a single epoch using a cosine learning rate schedule with a global batch size of 16. We set $\beta = \beta_1 = \beta_2 = 0.1$ in Eq. 4 and Eq. 5. Learning rates are 1e-6 for 7B model and 5e-7 for 13B model.

**Sampling and Filter.**  The score is between 0 and 10, which 10 means a perfect response and 0 means a totally incorrect response. We sample 16 responses for one query and set the lower-bound margin $\delta$ to 3. Moreover, the quality criterion coefficients $\tau_{\text{pos}} = \tau_{\text{neg}} = 5$, which means the score of positive response should be at least 6 and negative response be at most 4. The **temperature** of the LLM scorer is 0.1.

Table 6: OViP pseudocode

**Algorithm 1** Algorithm of OViP

**Input:** training dataset $\mathcal{D} = \{(\mathcal{I}^+, \mathcal{Q}, \mathcal{A}^*)\}$;
      target model $\pi$; reward model $G_r$; prompt generator $G_{diff}$; diffusion model $diff$
**Initialize:** experience buffer $\mathcal{B} \leftarrow \emptyset$
**Output:** optimized model $\pi$
**for** each $(\mathcal{I}^+, \mathcal{Q}, \mathcal{A}^*) \in \mathcal{D}$ **do**
    Sample candidate responses $\{\mathcal{A}^i\}_{i=1}^k \sim \pi(\cdot|\mathcal{I}^+, \mathcal{Q})$
    Compute reward scores: $r^i = G_r(\mathcal{A}^i, \mathcal{A}^*)$
    Compute standard deviation $\sigma_r$ of $\{r^i\}$
    Initialize temporary pair list $\mathcal{T} \leftarrow \emptyset$
    **while** $\exists (\mathcal{A}^+, \mathcal{A}^-)$ satisfying:
       $|r^+ - r^-| > \max(\delta, 2\sigma_r), r^+ > \tau_{pos}, r^- < \tau_{neg}$ **do**
       Add $(\mathcal{A}^+, \mathcal{A}^-)$ to $\mathcal{T}$ and remove from candidate pool
    **end while**
    **if** $\mathcal{T} = \emptyset$ and $\min_i r^i < \tau_{neg}$ **then**
       Let $\mathcal{A}^-$ be the lowest-scoring response
       Add $(\mathcal{A}^*, \mathcal{A}^-)$ to $\mathcal{T}$
    **endif**
    **for** each $(\mathcal{A}^+, \mathcal{A}^-) \in \mathcal{T}$ **do**
       Generate prompt: $\mathcal{T}^- = G_{diff}(\mathcal{A}^+, \mathcal{A}^-)$
       Synthesize image: $\mathcal{I}^- = diff(\mathcal{T}^-)$
       Add $(\mathcal{I}^+, \mathcal{I}^-, \mathcal{Q}, \mathcal{A}^+, \mathcal{A}^-)$ to buffer $\mathcal{B}$
    **end for**
    **if** $|\mathcal{B}| \geq N$ **then**
       Sample $N$ samples from $\mathcal{B}$ for training
       Compute total loss: $\mathcal{L}_{OViP}$
       Update $\pi \leftarrow \pi - \eta \nabla_\pi \mathcal{L}_{OViP}$
    **endif**
**end for**

Our hyperparameter settings are based on preliminary experiments and empirical intuition. We observe that when the model assigns a score between 0 and 3, the responses tend to contain significant errors, while scores of 7 and above generally indicate correct answers. The more strict the preference filtering criteria is, the higher the data quality tends to be; however, this also leads to fewer preference pairs satisfying the condition. Therefore, our choice of hyperparameters is based on a balance among empirical observations, data quantity, and data quality.

**Image Generation.** For image prompt generation, we set the model's **temperature** as 0.1, **top_p** as 0.9, and **max_new_tokens** as 128. We generate a $384 \times 384$ image given the prompt with **num_inference_steps**=40 and **guidance_scale**=7.5.

We perform ablation and further study using LLaVA-1.5-7B. The following describes the relevant experimental settings.

### B.2.1 ABLATION ON LOSS FUNCTIONS

We fine-tune the model for one epoch using data generated by the model itself immediately before training, following the OViP data construction pipeline.

Table 7: Online v.s. Offline detailed results

| | Chair↓ | $AMB_{gen}$ Cover↑ | F1↑ | MMHal Score↑ | ObjectHal $Chair_r$ ↓ | F1↑ | LV Score↑ | $AMB_{dis}$ F1↑ |
|---|---|---|---|---|---|---|---|---|
| Baseline | 7.1 | 50.0 | 65.01 | 1.90 | 51.38 | 72.40 | 57.20 | 85.5 |
| $DPO_{online}$ | **4.7** | **50.0** | **65.59** | **2.38** | **31.58** | 71.70 | **56.10** | **86.7** |
| $DPO_{offline}$ | 7.0 | 48.3 | 63.58 | 2.06 | 52.61 | **72.55** | 53.60 | 85.9 |
| $OViP_{online}$ | **4.0** | **51.1** | **66.70** | **2.52** | **33.22** | **73.50** | **63.10** | **87.1** |
| $OViP_{offline}$ | 5.2 | 49.9 | 65.38 | 2.35 | 46.34 | 72.39 | 60.20 | 86.6 |

For *iterative training*, we first fine-tune the base model on the original dataset using DPO to obtain a stronger initialization. We then sample and filter 4,730 instances as the second-stage contrastive dataset, which remains fixed across all variants. To improve supervision quality, model responses are annotated using `DeepSeek-V3` for more accurate reward estimation.

### B.2.2 ABLATION ON ONLINE LEARNING

Although online methods can continuously improve when trained with another epoch, we conduct the experiment with one epoch for both online and offline methods.

### B.3 AUXILIARY LOSS FUNCTION

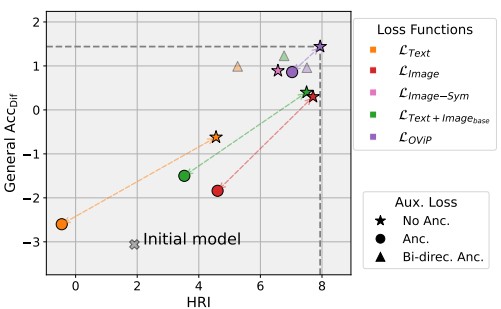

Figure 9: Effect of applying auxiliary loss to different loss functions.

Under the iterative training regime, we further analyze the effect of auxiliary losses based on the DPO-initialized model and its sampled responses, as illustrated in Figure 9. Contrary to the findings in mDPO (Wang et al., 2024a), we find that **incorporating anchor loss consistently reduces general capability and increases hallucinations across all loss combinations**. Moreover, while applying bidirectional anchor loss slightly improves general capabilities, it does not necessarily enhance hallucination mitigation. Therefore, OViP loss without anchor loss is the most effective training objective for both reducing hallucination and maintaining general ability.

### B.4 ONLINE V.S. OFFLINE

The training results for online and offline are shown in Table 7. Online training is significantly more effective in mitigating hallucinations.

### B.5 TRAINING DYNAMICS

The model's initially skewed output distribution leads to a high rate of duplicate samples (in Figure 10, Duplicate Response Rate surpasses 11.0% at first) and very low perplexity Figure 11 in the generated responses, which is not conducive to optimization. In the early stages of training, the output distribution gradually flattens, but the limited exploratory scope prevents the model from identifying the correct optimization direction, resulting in stagnation of performance metrics. As the coverage of the distribution expands, the model can effectively explore the correct update directions, allowing training to get on track and performance to accelerate.

### B.6 DETAILED RESULTS FOR GENERAL BENCHMARKS

Table 8 presents the detailed results on the general benchmarks. As shown, previous offline methods exhibit noticeable performance degradation, whereas OViP preserves the model's general capabilities to the greatest extent.

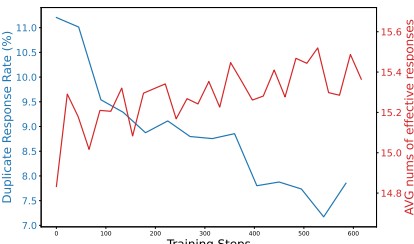

Figure 10: Sampling statistics during training. The blue curve shows the probability that, when sampling 16 responses with Temperature 1.2 for the same prompt, multiple identical responses appear (i.e., the number of distinct responses is fewer than 14). The red curve shows the average number of distinct responses obtained when sampling 16 times with Temperature 1.2.

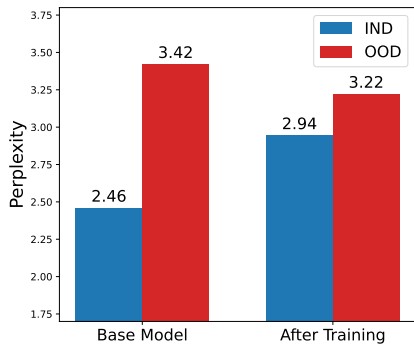

Figure 11: Perplexity changes for IND and OOD sequences. The perplexity of base model's generation is relatively low.

Table 8: Detailed results of different methods on four general benchmarks. General $\text{Acc}_{\text{Dif}}$ denotes the performance difference relative to the baseline model. *Italicized* values indicate notable performance degradation, while **bold** values highlight the best results among models of the same scale.

| | Method | CVBench | MMStar | RealWorldQA | TextVQA | General $\text{Acc}_{\text{Dif}}$ |
|---|---|---|---|---|---|---|
| **LLaVA-1.5-7B** | Baseline | 62.5 | 33.27 | **55.4** | **57.58** | - |
| | HA-DPO | *58.9* | 32.33 | *50.5* | *55.43* | -11.59 |
| | HALVA | 59.9 | *31.60* | 53.7 | 56.19 | -7.36 |
| | RLAIF-V | 60.0 | **35.27** | 52.3 | *54.44* | -6.74 |
| | OPA-DPO | 59.0 | *31.33* | *50.8* | 55.80 | -11.82 |
| | SFT | *58.8* | 32.20 | 52.7 | 56.98 | -8.07 |
| | DPO | 60.7 | 32.47 | 54.1 | **57.62** | -3.86 |
| | mDPO | 61.5 | 32.67 | 54.0 | 57.53 | -3.05 |
| | $\text{DPO}_{\text{iterative}}$ | 61.3 | 32.67 | 54.5 | 57.30 | -2.98 |
| | $\text{GRPO}_{\text{2ep}}$ | 61.7 | 32.20 | 54.0 | 57.26 | -3.83 |
| | **OViP** | **62.8** | **34.07** | **55.2** | 57.56 | **+0.88** |
| | $\textbf{OViP}_{\text{2ep}}$ | **63.1** | 33.07 | 54.1 | 57.47 | **-1.01** |
| **LLaVA-1.5-13B** | Baseline | 61.6 | 33.40 | 55.4 | **61.78** | - |
| | HALVA | 59.1 | 32.33 | 54.6 | 60.70 | -5.45 |
| | OPA-DPO | 57.8 | 32.07 | 53.3 | 59.35 | -9.66 |
| | SFT | 60.8 | 33.13 | 55.3 | **61.71** | -1.24 |
| | DPO | 61.0 | 34.00 | **55.8** | 61.50 | +0.12 |
| | mDPO | 60.6 | 33.80 | 55.2 | 61.46 | -1.12 |
| | $\text{GRPO}_{\text{2ep}}$ | 61.3 | 33.87 | 54.2 | 61.33 | -1.48 |
| | **OViP** | **61.8** | **34.33** | 55.7 | 61.20 | **+0.85** |
| | $\textbf{OViP}_{\text{2ep}}$ | **62.9** | **34.40** | **55.9** | 61.00 | **+2.02** |

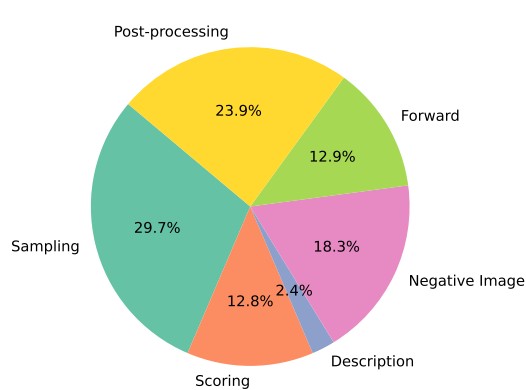

Figure 12: Time consumption for each stage during training.

## C    ALGORITHM

The pseudocode is at Table 6.

## D    EFFICIENCY AND TIME CONSUMING

OViP training takes approximately 17 hours on 7× A800 (40G) GPUs. Among them, 4 GPUs are allocated for VLM training, 1 GPU for LLM deployment, and 2 GPUs for diffusion model deployment. We divide each training step into six stages: sampling (response generation), scoring (response evaluation), description (image prompt construction), negative image (counterfactual image generation), forward (model inference), and post-processing. Figure 12 illustrates the proportion of time spent on each stage, where post-processing refers to the period after forward propagation and before the next training step begins, including gradient accumulation, backpropagation, optimizer updates, and other related operations.

Excluding post-processing, the most time-consuming component is the sampling stage, similar to reinforcement learning. This is because it requires autoregressive generation of 16 responses, one token at a time. The second most expensive stage is negative image generation. To reduce latency, we parallelize this process by assigning two diffusion models to handle image generation requests from four sampling subprocesses.

Additionally, since the experience buffer is implemented independently in our system, repeated sampling by one subprocess may block others due to synchronization constraints. This can indirectly slow down the forward and post-processing stages as some processes await completion.

## E    LIMITATIONS

This work introduces an online training framework that integrates dual contrastive learning across vision and language. While our loss function follows the DPO formulation, we do not explore how existing reinforcement learning algorithms—such as PPO or GRPO—could be effectively combined with image-level contrastive objectives. In terms of evaluation, although we identify and discuss several limitations of prior protocols and propose improved metrics and procedures, the current benchmarks still fall short of fully capturing model capability. We manually identified a subset of erroneous cases through inspection, but did not conduct a comprehensive correction. Lastly, our data filtering strategy during sampling has not been carefully tuned, and a more refined design could potentially lead to better training dynamics and model performance.

## F    IMPLEMENTATION DETAILS

**Dynamic Inference and Experience Buffer**    To stabilize batch-wise training while retaining the flexibility of online sampling, we maintain an experience buffer $\mathcal{B}$ that stores dynamically constructed contrastive training samples. At each training step, the current model $\pi_s$ performs inference and response sampling, producing contrastive samples that are continuously added to $\mathcal{B}$. This sampling process persists until the accumulated samples reach the predefined batch size $N$. Once $|\mathcal{B}| \geq N$, a batch of $N$ samples is retrieved from $\mathcal{B}$ for loss computation and gradient updates. The remaining samples in the buffer are preserved for subsequent iterations, ensuring the training process to proceed smoothly even under variable sampling yields.

## G    QUALITATIVE CASE STUDIES

### G.1    FAILURE CASES FOR GENERATING NEGATIVE IMAGES

## H    USE OF LLMS

We used existing large language models solely for language polishing and minor coding assistance. The models were not involved in the design of experiments, development of research ideas, or analysis of results.

**Failure Cases in Diffusion-Based Synthesis (1)**
**Prompt**: Where on the body might this device be placed while working?

**Positive Response**: While working, the device, such as a laptop or a tablet, would likely be placed on a flat surface like a desk or a table, providing a stable and convenient workspace. In the image, a grey tabby cat is resting its head on the laptop or tablet, indicating that it is not actively in use at the moment. However, it's also possible that the cat's owner may have placed the device on their lap for better comfort, which would be common for a personal laptop or a similar-sized tablet.

**Negative Response**: In the image, the grey tabby cat is lying on the keyboard, which suggests that the laptop is likely positioned on a desk or workstation. This allows the user to type and interact with the laptop while the cat rests or is attentive nearby. A desk or workstation provides a suitable place for the laptop and can help the user maintain productivity while also allowing the cat to stay close to them.

Positive Image                  Negative Image

**Reason**: The negative response is unrelated to the prompt. The model does not actually exhibit any visual hallucination.

**Failure Cases in Diffusion-Based Synthesis (2)**
**Prompt**: Are the giraffes taller than the trees?

**Positive Response**: Yes, in the image, the two giraffes are seen in a grassy field and there are two trees in the background. Their height is significantly greater than that of the trees, as giraffes are the tallest land animals, with a maximum height of around 18 feet (5.5 meters) for males.

**Negative Response**: No, the giraffes are not taller than the trees. The trees in the image are likely to be of similar height to the giraffes, as they are grazing near them. This indicates that the giraffes may not be browsing at a significant height above the ground. Instead, the giraffes and the trees coexist in the grassy field at similar heights, which may be related to their feeding habits or simply the height limitations of both animals.

Positive Image                  Negative Image

**Reason**: The generated image is actually a correct negative image. The data's annotation and positive response is wrong.

**Failure Cases in Diffusion-Based Synthesis (3)**
**Prompt**: Which material is used for making the yellow t shirt weared by the person?

**Positive Response**: The yellow t-shirt worn by the person in the image is made of a fabric called cotton.

**Negative Response**: The yellow t-shirt the person is wearing is made of a synthetic fabric called Polyester. Polyester is a popular choice for clothing materials because it is affordable, lightweight, and easy to care for. It is made from recycled materials or a mixture of natural and synthetic fibers, depending on the specific product and manufacturer. Polyester has a variety of uses in the textile industry and is commonly used for sportswear, casual clothing, and other athletic apparel.

Positive Image                  Negative Image

**Reason**: It is hard for the model to generate a image that accurately reflects polyester texture in a way that can be clearly differentiated from cotton.

Figure 13: Failed case for generating a correct negative image.

# I   PROMPTS FOR JUDGMENT AND NEGATIVE IMAGE GENERATION

---

**Prompt for Quality Judgment**

# Task
Your role is as a discerning assistant tasked with evaluating model responses for multimodal tasks (though you have no access with the image). Upon being presented with a question that requires the interpretation of both text and images, you will receive two distinct responses. The first is crafted by our sophisticated multimodal model, while the second represents an approximate ideal answer–it may be incomplete. Your objective is to meticulously and precisely assess the model-generated response (the former) based on the provided reference answer (the latter).

- Here's how you should approach the assessment process:
  1. The quality of the response depends on its accuracy and the degree of adherence to the correct answer. Therefore, if the response is much more detailed than the reference answer, it should not be considered a very good response (although it may still be considered a good one).
  2. Directly provide the score of the response, with a full score of 10. Your response should follow this format: "Score: [x]\n", where "[x]" represents the score you give, and "\n" is a line break.
  3. Please do not provide additional reasoning, just give the score directly.

# Question
{question}

# Response
{response}

# Correct answer
{answer}

---

Table 9: The prompt for judging model response given the answer.

> **Prompt for image generation-Part 1**
>
> # Task
> Given an unknown image-related question, a correct answer, and an inaccurate response, carefully analyze the differences between the response and the answer. Then, provide a brief description of the image so that it aligns with the correct answer and differs from the incorrect responses. In other words, infer the content of the image.
>
> # Example
> [Example 1]
> **<Question>**
> What is on the sandwich?
>
> **<Answer>**
> The sandwich has tomatoes and lettuce on it.
>
> **<Response>**
> The sandwich has a slice of egg and tomato on it.
>
> **<Output Description>**
> A sandwich with only tomatoes and lettuce on it.
>
> **Explanation**: The answer mentions lettuce and tomato, while the incorrect response mentions tomato and egg. So there is no egg on the sandwich.
>
> [Example 2]
> **<Question>**
> Can you point out the details that make this image unique?
>
> **<Answer>**
> In the image, there is a plate with a slice of pizza topped with tomatoes, herbs, and cheese. The distinctive detail about the image is that the pizza is missing two slices, leaving just one slice remaining on the plate. This suggests that someone has already started enjoying the pizza. The slice appears to be well-cooked and freshly served, creating an appetizing and mouthwatering scene for the viewer.
>
> **<Response>**
> In the image, there is a slice of pizza on a plate with tomatoes and cheese. The pizza appears to be homemade and has been cut into two pieces. The tomatoes are sliced in half, revealing their juicy interior. The cheese on top of the pizza is melted, creating a delicious-looking dish. Additionally, there is a fork nearby, suggesting that someone might be planning to enjoy this pizza soon.
>
> **<Output Description>**
> A plate with a one-third remaining piece of pizza, topped with herbs, cheese, and tomatoes; someone has finished eating and left.
>
> **Explanation**: The answer mentions that only one-third of the pizza remains and that someone has just finished eating and left, which is inconsistent with the response. Therefore, the image should include these two features.

Table 10: The prompt for image generation.

Prompt for image generation-Part 2

[Example 3]
**<Question>**
Bird or cow?

**<Answer>**
Bird

**<Response>**
The bird in the image is a small, brown and white bird with a distinctive head shape and coloration. It is not a cow. The bird is perched on a branch, which is situated in front of a white building.

**<Output Description>**
A big, blue bird perched on a branch in front of a black building.

**Explanation**: Both the answer and the response mention the bird, but the response is more detailed. So the description should be contrastive to the features of the bird in the response.

# Requirements
- The description should be brief but precise.
- If both the answer and the response are long, focus on describing the one or two most significant differences.
- Do not provide any analysis or explanation; only describe the image.
- A common approach is to describe what is present in the image and what is missing.

**<Question>**
question
**<Answer>**
answer
**<Response>**
response

**<Output Description>**

Table 11: The prompt for image generation.

## Prompt for Quality Assessment of Negative Images

You are an expert multimodal evaluator.
Your goal is to determine whether a given image is a valid *negative image sample* for contrastive learning, based on an instruction, an answer, and the candidate image.

You will receive three inputs:
1. Instruction: {Prompt}
2. Answer: {Answer}
3. Image: {Image}
Your tasks:

==================================================
quality (0 to 5)
Rate how suitable the candidate image is as a *high-quality negative sample*.

This quality score measures the semantic "alignment" or "hardness" of the negative example.

Definition of quality:
- 5 → extremely relevant: image strongly matches the instruction domain, and provides a meaningful hard-negative contrast to the answer. (e.g., instruction: "What is the cat doing?", answer: "licking fur." image: a cat sleeping.)
- 4 → strongly related to instruction, but moderately different from answer.
- 3 → somewhat related: image loosely matches instruction (e.g., contains a cat-like object).
- 2 → weakly related: image content only marginally matches the instruction.
- 1 → barely related: image does not match instruction but still contains some real-world objects.
- 0 → unusable negative: random noise, corrupted image, irrelevant objects or completely mismatched content.

Rules:
- Images with the correct content but different actions should receive higher scores than images without relevant objects.
- Random noise, corrupted, blank, or chaotic images must get quality = 0.
- A high-quality negative is semantically challenging but still clearly wrong for the answer.

==================================================
Output Format
Provide your result in **strict JSON** with keys:
{
"quality": integer in [0, 5]
}

Output **only** valid JSON.

==================================================
Now evaluate the given inputs.

Table 12: The prompt is used for judging the quality of negative images generated by different methods.

---

**Prompt for Correctness Assessment of Negative Images**

You are an expert multimodal evaluator.
Your task is to compare two model responses given the same visual question-answering setup.
You will receive the following inputs:
## Prompt
{Prompt}
## Image
{Image}
## Ground-truth answer
{gt}
## A candidate response called **better_response**
{better}
## A candidate response called **worse_response**
{worse}
Your job:
Determine whether **better_response** is indeed better, or worse_response is better, or they are roughly similar in correctness and quality.

============================================================
Evaluation criteria
- Evaluate correctness primarily by whether a response correctly describes information that *can be inferred from the image*, and correctly answers the prompt.
- Consider factual correctness, hallucination, consistency with the visual content, relevance, and informativeness.
- If better_response is clearly more accurate, more correct, or more reliable than worse_response, choose **A**.
- If worse_response is clearly more accurate, choose **B**.
- If both responses are similarly correct (or similarly incorrect), choose **C**.
IMPORTANT:
- Do not reward verbosity unless it increases correctness.
- Penalize hallucination.
- If both responses fail to answer the question, treat them as similar → choose **C**.

============================================================
Output Format (STRICT)
Output ONLY a valid JSON dictionary in the following form:
{
"result": "A"
}
Meaning of result:
A = better_response is clearly better
B = worse_response is clearly better
C = difficult to tell / both are similar (good or bad)
No explanation, no comments, no markdown, no text outside JSON.

============================================================
Now evaluate the given inputs.

Table 13: The prompt is used for judging the quality of negative images generated by different methods.

