# OpenReview forum: "OViP: Online Vision-Language Preference Learning for VLM Hallucination"
_ICLR.cc/2026/Conference — ICLR 2026 Conference Withdrawn Submission_

### Official Review · Reviewer_kH4S · 2025-11-01

**Soundness:** 2
**Presentation:** 3
**Contribution:** 2
**Rating:** 4
**Confidence:** 4

**Summary:**

OViP is an online vision-language preference learning framework designed to reduce hallucinations in large vision-language models (LVLMs), which often invent objects, attributes, or spatial relations that are not actually in the image. It continuously samples the model’s own responses during training, identifies good vs. bad answers, and then uses an external LLM plus a diffusion model to synthesize targeted “negative” images and contrastive preference pairs that reflect the model’s real failure modes instead of relying on static or randomly edited data. OViP jointly optimizes both the text side (the response) and the image side, providing real-time supervision that pushes the model to stay faithful to what’s in the image rather than overfitting to language priors. Experiments show that OViP lowers hallucination rates, preserves general multimodal ability, improves training efficiency compared to prior offline DPO-style or online RL-style methods, and also highlights that evaluation should balance hallucination suppression with informativeness.

**Strengths:**

1. The method builds preference data online from the model’s own mistakes, instead of relying only on fixed, offline edits. This directly targets the model’s real hallucination modes during training and can make the supervision more relevant and efficient.

2. It jointly optimizes both text faithfulness and visual grounding, aiming to reduce hallucination without making the model overly timid or uninformative.

**Weaknesses:**

1. The novelty needs to be clarified: similar on-policy training ideas already exist. For example, SIMA[1] also scores the model’s own generated samples and uses them as positive/negative supervision. The paper should explain more clearly what is fundamentally new beyond that prior works.

2. The approach depends on synthetic images from a generative model. But current image generators are not perfectly reliable, so the “negative” images might themselves be noisy or wrong. The paper should justify why supervision built on these generated images can be trusted.

3. The experiments should include stronger, more recent LVLM baselines (e.g., Qwen 2.5 VL series) to demonstrate that the method still helps on modern state-of-the-art models, not just older ones.

[1] Enhancing Visual-Language Modality Alignment in Large Vision Language Models via Self-Improvement.

**Questions:**

Please refer to the weaknesses part.

---

> ### Author Response · Authors · 2025-11-27
>
> We thank you for your review. Here is our response to your concerns.
> > Weakness 1: Novelty
>
> We have provided a detailed response on the novelty and key contributions in the public comment.
>
> > Weakness 2: Probably Unreliable Negative images
>
> We provide an analysis of the quality of the synthetic images, which shows that, compared with previous model-free negative image generation approaches, our method achieves higher visual quality while maintaining a low error rate, making it more suitable as hard negatives for contrastive learning. In addition, we also demonstrate this observation experimentally.
>
> Quality of the synthesized negative images
> | Method | Error rate | Quality | Cosine Similarity to the positive image |
> |-----|-----|-----|-----|
> | Model-based Synthesis(Ours) | 9.41% | 4.20 | 0.6224 |
> | Random Cropping | 70.86% | 3.94 | 0.9672 |
> | Random Image    | 3.39% | 2.52 | 0.4177 |
>
> Results for image synthesis methods
> | Method | AMB_Chair  |   AMB_Cover  |   *AMB_F1*   |  *MMHAL*  |  OBJ_Cha  |  *OBJ_F1*   |   *LV*   | *AMB_Discriminative*  | *HRI* |  *General* |
> | ----- | ----- | ----- | ----- | ----- | ----- | ----- | ----- | ----- | ----- | ----- |
> | Model-based Synthesis (Ours)  | 3.30 | 52.80 | **68.30** | 2.70 | 28.21 | **74.14** | **63.60** | 85.70 | **10.88** | **+1.44** |
> | Random Cropping | 3.50 | 52.50 | 68.00 | 2.50 | 27.21 | 73.75 | 62.50 | 86.40 | 9.76 | +0.34 |
> | Random Image | 3.40 | 50.50 | 66.33 | **3.37** | 24.20 | 73.11 | 61.20 | **86.60** | 9.29 | -0.66 |
>
> Constructing high-quality hard negative images online yields stronger performance than prior model-free negative construction approaches. In particular, the synthetic negative generation strategy used in OViP achieves superior results and preserves the model’s general capabilities more effectively.
>
> OViP consistently samples from model's own hallucination distribution while offline methods can not guarantee the derived images truly lie in hallucination spaces. Although diffusion-based synthesis may introduce certain implicit biases, our experiment shows that the advantages it provides for generating hallucination-targeted negative images largely outweigh these biases.
>
>
>
> > Weakness 3: Experiments on Qwen2.5-VL
>
> Thank you for the advice. We run additional experiments on Qwen2.5-3B, and the results show that OViP delivers the best performance.
>
> |   Method | AMB_Discriminative | AMB_Chair| AMB_F1| LV |  MMHal   | OBJ Chair$_r$  |  OBJ F1  |  General Acc$_{\mathrm{Dif}}$ |
> | ---- | ---- | ---- | ---- | ---- | ---- | ---- | ---- | ---- |
> | Qwen2.5 | 91.60 | 6.60 | 74.96 | 87.20 | 3.45 | 19.27 | 68.70 | - |
> | DPO | 91.50 | 8.10 | 73.64 | 88.20 | 3.87 | 16.61 | 65.84 | -4.28 |
> | SFT | 91.50 | 4.80 | 71.31 | 75.50 | 3.44 | 23.61 | **72.75** | +0.73 |
> | mDPO | 91.50 | 8.30 | **77.49** | **88.50** | **3.97** | 15.44 | 66.38 | -3.31 |
> | **OViP** | 91.50 | 4.50 | **75.23** | **88.50** | **4.33** | 15.60 | **73.44** | +1.47 |

---

### Official Review · Reviewer_zfn5 · 2025-11-01

**Soundness:** 3
**Presentation:** 3
**Contribution:** 2
**Rating:** 4
**Confidence:** 4

**Summary:**

In this paper, the authors propose an online preference tuning method to reduce hallucination in LVLMs by using diffusion models to generate paired images.

**Strengths:**

The experiments are solid, and the writing is clear.

**Weaknesses:**

1. The method doesn't have particularly obvious novelty, as similar ideas have been explored before in V-DPO.
2. Missing fine-grained results on general benchmarks; experimental results in this area need to be supplemented.
3. The architecture is limited to LLaVA, and this model is relatively old. It's unclear whether this method would work on newer LVLM architectures.

**Questions:**

The model could be updated to newer models and other architectures, such as the Qwen-VL series, to validate the generalizability of the method.

What is the quality of the negative samples generated by the diffusion model? How does it compare to other methods, such as image editing?

---

> ### Author Response · Authors · 2025-11-27
>
> > Weakness 1: Novelty
>
> We have provided a detailed response on the novelty and key contributions in the public comment.
>
> > Weakness 2: Miss Detailed Results
>
> Thanks for your suggestion, we will provide fine-grained evaluation results in Appendix B.5, Table 6.
>
> Previous offline methods exhibit noticeable performance degradation, whereas OViP preserves the model’s general capabilities to the greatest extent.
>
> > Weakness 3 & Question 1: Need for Newer LVLM Architectures
>
> We followed the OPA-DPO settings to enable a fair and direct comparison. LLaVA-1.5 is one of the most widely adopted models in VLM hallucination research. That said, we are also open to extending our method to a broader range of models like Qwen-Series.
>
> |   Method | AMB_Discriminative | AMB_Chair| AMB_F1| lv |  MMHal   | OBJ Chair$_r$  |  OBJ F1  |  General Acc$_{\mathrm{Dif}}$ |
> | ---- | ---- | ---- | ---- | ---- | ---- | ---- | ---- | ---- |
> | Qwen2.5-3B | 91.60 | 6.60 | 74.96 | 87.20 | 3.45 | 19.27 | 68.70 | - |
> | DPO | 91.50 | 8.10 | 73.64 | 88.20 | 3.87 | 16.61 | 65.84 | -4.28 |
> | SFT | 91.50 | 4.80 | 71.31 | 75.50 | 3.44 | 23.61 | **72.75** | +0.73 |
> | mDPO | 91.50 | 8.30 | **77.49** | **88.50** | **3.97** | 15.44 | 66.38 | -3.31 |
> | **OViP** | 91.50 | 4.50 | **75.23** | **88.50** | **4.33** | 15.60 | **73.44** | +1.47 |
>
> > Question 2: Quality of the negative samples
>
>  Thanks for your question, we provide a diagnostic analysis of synthesized negative images here:
> Quality of the synthesized negative images
> | Method | Error rate | Quality | Cosine Similarity to the positive image |
> |-----|-----|-----|-----|
> | Model-based Synthesis(Ours) | 9.41% | 4.20 | 0.6224 |
> | Random Cropping | 70.86% | 3.94 | 0.9672 |
> | Random Image    | 3.39% | 2.52 | 0.4177 |
>
> Regarding the the image editting method, in practice, we observe that this produce lower quality results compared to directly generating images from prompts. (We will provide some examples in the Appendix)

---

### Official Review · Reviewer_R4wf · 2025-11-04

**Soundness:** 2
**Presentation:** 3
**Contribution:** 3
**Rating:** 4
**Confidence:** 2

**Summary:**

This paper focuses on the problem of hallucinations generated by large visual language models (LVLMs) under visual input conditions. The paper argue that although there are methods that utilize preference learning (such as DPO) to alleviate hallucinations, these methods often rely on pre-defined or randomly edited negative samples that do not match the distribution of the model's true failure patterns, thereby limiting the training effectiveness. To this end, the paper proposes the Online Vision language Preference Learning (OViP) framework, dynamically identifying "good/bad" pairs from candidate answers generated by the model during the training process, and using a diffusion model to generate negative images to construct visual-language pairs. In this way, OViP constructs preference learning signals on both the text and image sides, achieving joint adjustment of text preferences and visual preferences in the model output. The experimental results show that OViP can reduce the occurrence of hallucinations on multiple hallucination detection and universal visual language task benchmarks, and the training efficiency is better than previous methods. Overall, this work proposes an online, dynamic negative sample generation mechanism that is triggered by the model's own errors, providing a training scheme that is more in line with the actual failure distribution for illusion relief.

**Strengths:**

- This paper addresses a crucial problem in VLMs: hallucinations. The paper propose OViP, which introduces the concepts of "online construction of negative samples" and "joint image-text preference learning," demonstrating good performance on some datasets.

**Weaknesses:**

- The method relies on generative models (LLM evaluation, diffusion model for generating negative images)—these steps increase method complexity. The quality of the generated negative images and whether the synthesized negative samples can truly cover the illusion space may limit generalization. Although the authors provide an efficiency analysis, their robustness in large-scale/diverse scenarios is not yet fully demonstrated.
- The core components of this method (preference learning, negative sample generation, and negative image synthesis) already have conceptual or approximate aspects in existing research. While the paper includes a review of relevant work, it does not highlight the core breakthroughs of this method compared to the closest comparable approaches. Furthermore, the underlying mechanisms of core technologies such as LLM evaluation and negative sample image generation are not analyzed in depth.
- Some Figures are difficult to understand intuitively, such as the two left images in Figure 1, which lack appropriate legends and explanations, making it difficult for readers to quickly understand the content that the images are trying to convey.

**Questions:**

- Could the authors provide more details and analysis of the negative image generation mechanism, such as the diversity/coverage of generated image, examples of failed samples, and whether there is a risk that "generating incorrect images" might actually teach the model to "escape the real task distribution"?
- Could the author add more "mechanistic discussion" to the methodology section.  A deeper explanation of the design principles of the core methods, rather than just a method description? For example, why is the proposed LLM-based method for generating positive and negative response pairs better?

---

> ### Author Response · Authors · 2025-11-27
> **Response to the reviewer's concerns (Part 1)**
>
> We thank you for the detailed review. Below is our response to your concerns.
> > Weakness 1: Complexity, coverage and generalization
>
> ### Regarding method complexity.
>  Although our framework employs an LLM‐based evaluator and a diffusion model to generate negative samples, we argue that this does not substantially increase practical complexity. Implementation-wise, these components are deployed as lightweight inference services via FastAPI, and training simply interacts with them through standardized API calls, without modifying model architecture or drastically changing training flow. Conceptually, these models only serve as auxiliary supervisors rather than forming additional learning modules, and thus introduce minimal engineering overhead. Importantly, our experiments show that incorporating these two components leads to significantly higher training efficiency and larger performance gains compared to alternative designs, suggesting that the additional complexity yields considerable benefits (Figure 4).
> ### Regarding the coverage and reliability of synthesized negative samples.
>  A key property of OViP is that **negative samples are directly driven by the model’s own hallucinated outputs, and thus are drawn from the model’s evolving hallucination distribution** instead of synthetic or heuristic perturbations. We also observe consistent improvement across training epochs (Table 1, Figure 5), indicating stable effectiveness and generalization of the synthesized samples. In contrast, prior model-free negative construction strategies (random cropping, random noise, or random real images) and even model-based but non-hallucination-guided approaches (V-DPO, diffusion-based editing) do not sample directly from hallucination space and therefore offer limited adaptability. Our approach is inherently more aligned with the failure modes the LVLM actually exhibits during training, enabling more robust and sustained gains.
>
>
> > Weakness 2: Novelty and lack of deeper analysis
> ### Regarding novelty.
> We have provided a detailed response on the novelty and key contributions in the public comment.
>
> ### Regarding the need for more mechanistic discussion.
>  We thank the reviewer for pointing this out. While our main focus is the effectiveness of the proposed online vision-language preference learning paradigm, we agree that a deeper analysis of key components will strengthen the work. Accordingly, we provide: (1) an assessment of LLM-based evaluation reliability, (2) a diagnostic analysis of synthesized negative images, and (3) ablations examining the effect of negative-image construction strategies. These additions provide clearer insights into why and how each component contributes to overall performance.
>
> 1. LLM evaluation reliability
>
> We employ GPT as an automatic evaluator to assess the filtered sample pairs, and manually verify the cases where GPT judges the pair as “incorrect.” Based on this procedure, we quantify the scoring bias of the LLM. About **3.3%** of the pairs have unclear or incorrectt preferences. Overall, the results show that providing the correct answer and letting the LLM automatically determine the correctness of model responses is a reliable strategy.
>
> 2. Quality of the synthesized negative images
>
> | Method | Error rate | Quality | Cosine Similarity to the positive image |
> |-----|-----|-----|-----|
> | Model-based Synthesis(Ours) | 9.41% | 4.20 | 0.6224 |
> | Random Cropping | 70.86% | 3.94 | 0.9672 |
> | Random Image    | 3.39% | 2.52 | 0.4177 |
>
> 3. Experimental results for image synthesis methods
>
> | Method | AMB_Chair  |   AMB_Cover  |   *AMB_F1*   |  *MMHAL*  |  OBJ_Cha  |  *OBJ_F1*   |   *LV*   | *AMB_Discriminative*  | *HRI* |  *General* |
> | ----- | ----- | ----- | ----- | ----- | ----- | ----- | ----- | ----- | ----- | ----- |
> | Model-based Synthesis (Ours)  | 3.30 | 52.80 | **68.30** | 2.70 | 28.21 | **74.14** | **63.60** | 85.70 | **10.88** | **+1.44** |
> | Random Cropping | 3.50 | 52.50 | 68.00 | 2.50 | 27.21 | 73.75 | 62.50 | 86.40 | 9.76 | +0.34 |
> | Random Image | 3.40 | 50.50 | 66.33 | **3.37** | 24.20 | 73.11 | 61.20 | **86.60** | 9.29 | -0.66 |
>
> Constructing high-quality hard negative images online yields stronger performance than prior model-free negative construction approaches. In particular, the synthetic negative generation strategy used in OViP achieves superior results and preserves the model’s general capabilities more effectively. Moreover, enabling the model to generate negative samples that carry hallucination-related semantics can further lead to substantial performance improvements in training.

---

> > ### Author Response · Authors · 2025-11-27
> > **Response to the reviewer's concerns (Part 2)**
> >
> > > Weakness 3: Some Figures are not Clear.
> > We apologize for the confusion caused by the current presentation.
> > For Figure 1, the two diagrams on the left illustrate the fundamental difference between offline training and our OViP pipeline. In the offline setting, the model is updated based on a fixed, static dataset, and therefore model updates do not affect future training samples. In contrast, OViP forms a dynamic closed loop: the updated model continuously produces new samples, which are filtered, transformed (via LLM evaluation and diffusion-based synthesis), and immediately used for the next round of training.
> >
> > In addition, we guess that Figure 6 may also be difficult to interpret. To clarify, we collect outputs from the base model and various trained models, score them using the same evaluation protocol as in training, and track how the output-quality distribution shifts after training. Since the raw score distribution is discrete, we apply smoothing to present its evolution as a continuous curve.
> >
> > > Question 1-1: Failed examples
> >
> > Thank you for the suggestion. We will include them in the Appendix to improve readability.
> >
> > > Question 1-2: Risk of misalignment
> >
> > From the optimization perspective, misalignment only occurs when a negative sample is a false negative. In our setting, this would require the generated negative image to perfectly correspond to the positive answer. While in our evaluation (provided in Response Part 1), we demonstrate that compared with the model-free baselines (random images, random cropping), our approach achieves a much lower error rate while maintaining sufficient negative sample quality.
> >
> > > Question 2: Mechanistic Discussion & Design Principles
> > The design of OViP is modular. The core requirement of the framework is to provide real-time textual and visual preference signals without extra teacher models.
> >
> > **(1)** First, based on model-generated samples and offline ground-truth answers, the framework must identify positive and negative candidates. In practice, OViP adopts a widely-used LLM-as-a-judge and selects preference pairs using score-based filtering. This also allows us to avoid relying on a more powerful teacher model for real-time supervision.
> >
> > **(2)** Second, the framework needs to project textual hallucinations into the visual space. OViP leverages an LLM to derive contrastive prompts and uses a diffusion model to sample negative visual inputs accordingly.
> >
> > Given that model-free approaches are ineffective in this setting, we also experimented with image-editing based strategies for constructing negative images, and we identified two potential paths:
> >
> > - First detect entities and mask them, then modify the corresponding regions. However, this approach requires an additional masking model and depends on deep semantic understanding of images. We believe this introduces extra system complexity, and the masking task may exceed the visual understanding ability of the target VLM, raising risks similar to teacher-model distillation. Therefore, we did not adopt this direction.
> >
> > - Directly using an image-editing model to repaint the entire image. We have tried this method and observed that it produces lower quality results compared to directly generating images from prompts. (We will show this in the Appendix)
> >
> > Consequently, we adopted the current OViP strategy.
> >
> > **(3)** Third, the framework trains the LVLM using both textual and visual preference signals. We examined several objective formulations and selected the one that achieves the best empirical stability.
> >
> > At each stage, we balanced engineering complexity with practical effectiveness and integrated these components into a unified OViP framework.

---

### Author Response · Authors · 2025-11-27

We thank the reviewers for their feedback regarding novelty. Our main contribution lies in proposing a general framework that integrates online preference learning with visual counterexample construction, enabling LVLMs to learn from model-specific failures on both the response side and the image side.
Recent progress in mitigating hallucination has followed two largely independent directions.

- Some works explore visual preference optimization, demonstrating that training with paired images can reduce visually-grounded hallucinations. However, these approaches are typically implemented under an offline setting. Our analysis shows that such static supervision often leads to overfitting and degradation in informativeness and general LVLM capabilities. OViP instead performs visual preference learning in an online manner, achieving both stronger performance and higher training efficiency.
- On the other hand, a smaller set of studies like SIMA and OPA-DPO begins to explore online preference learning as a way to reduce hallucination in LVLMs. These efforts demonstrate that leveraging online supervision can be beneficial, though they only operate on the textual side. They operate like extensions of LLM preference training rather than addressing the multi-modal nature of hallucination. Our experiments demonstrate that incorporating image-side supervision online yields additional and complementary gains.

In addition, we improve the evaluation protocol and reveal that many prior offline methods induce substantial degradation in informativeness and general-purpose capabilities, leading to an overestimation of their gains (Section 3.3, Table 1). We further provide analyses suggesting why combining online learning with visual supervision can yield more reliable improvements (Section 4.3).

---

### Note · Authors · 2026-01-05

I have read and agree with the venue's withdrawal policy on behalf of myself and my co-authors.